# A highly active *Burkholderia* polyketoacyl-CoA thiolase for production of triacetic acid lactone

Zilong Wang[1,2,3,10], Seokjung Cheong[1,2,3,10], Jose Henrique Pereira[1,4], Weixi Hu[1,5,6], Yifan Guo [1,5,6], Andy DeGiovanni[1,4], Guangxu Lan [1,7], Jinho Kim[1,7], Robert W. Haushalter[1,7], Taek Soon Lee [1,7], Paul D. Adams [1,4,8] & Jay D. Keasling [1,2,7,9] ✉

Triacetic acid lactone (TAL) is a versatile platform chemical traditionally biosynthesized via decarboxylative Claisen condensation by 2-pyrone synthase. However, this route is limited by poor efficiency and dependence on malonyl-CoA. Here, we show that non-decarboxylative Claisen condensation by polyketoacyl-CoA thiolases offers a more efficient alternative. Through mining homologs of a previously reported enzyme from *Cupriavidus necator*, we identify five thiolases with TAL production activity. One candidate, BktBbr from *Burkholderia* sp. RF2-non_BP3, exhibits approximately 30-fold higher activity in vitro and supports 30-fold higher TAL titers in *Escherichia coli* compared to the original enzyme. Fed-batch fermentation achieves titers up to 2.8 g L$^{-1}$. Structural analysis of BktBbr co-crystallized with CoA esters guides rational engineering to further enhance performance. Our discovery of a highly active thiolase establishes an alternative enzymatic route to produce TAL efficiently, providing a scalable foundation for sustainable biomanufacturing.

Microbial biosynthesis of chemicals and fuels is an important research topic as it seeks to replace petroleum feedstocks with renewable biomass or one-carbon gasses, reduce the emissions of greenhouse gasses and pollutants, and improve flexibility and capital efficiency[1–3]. Enzymatic reactions constitute the biosynthetic pathways to those molecules, so the selection of enzymes with high activity and selectivity towards the target product is crucial to improve the titer, rate, and yield (TRY). The rise of bioinformatics and systems biology has offered more diverse tools for enzyme discovery and selection[2,4,5], and directed evolution and computational- and artificial intelligence-assisted

rational design can be used to further enhance the desired enzyme activity and performance[3,6–9].

Triacetic acid lactone (4-hydroxy-6-methyl-2-pyrone, TAL) is a platform chemical used in the synthesis of food preservatives and additives, fragrances, and a variety of other chemicals[10,11] and polymers[12–14]. As such, it has been categorized as a bio-privileged molecule, and it can be synthesized both chemically and biologically[15]. Plants such as *Gerbera hybrida* (Daisy) naturally produce TAL using acetyl-CoA as the initial precursor and two rounds of decarboxylative Claisen condensation with malonyl-CoA catalyzed by a type III poly-

[1]Joint BioEnergy Institute, Emeryville, CA, USA. [2]Department of Chemical and Biomolecular Engineering, University of California, Berkeley, CA, USA. [3]QB3 Institute, University of California, Berkeley, CA, USA. [4]Molecular Biophysics & Integrated Bioimaging Division, Lawrence Berkeley National Laboratory, Berkeley, CA, USA. [5]Department of Chemistry, University of California, Berkeley, Berkeley, CA, USA. [6]Department of Molecular and Cell Biology, University of California, Berkeley, Berkeley, CA, USA. [7]Biological Systems and Engineering Division, Lawrence Berkeley National Laboratory, Berkeley, CA, USA. [8]Department of Bioengineering, University of California, Berkeley, Berkeley, CA, USA. [9]Center for Biosustainability, Danish Technical University, Lyngby, Denmark. [10]These authors contributed equally: Zilong Wang, Seokjung Cheong. ✉e-mail: keasling@berkeley.edu

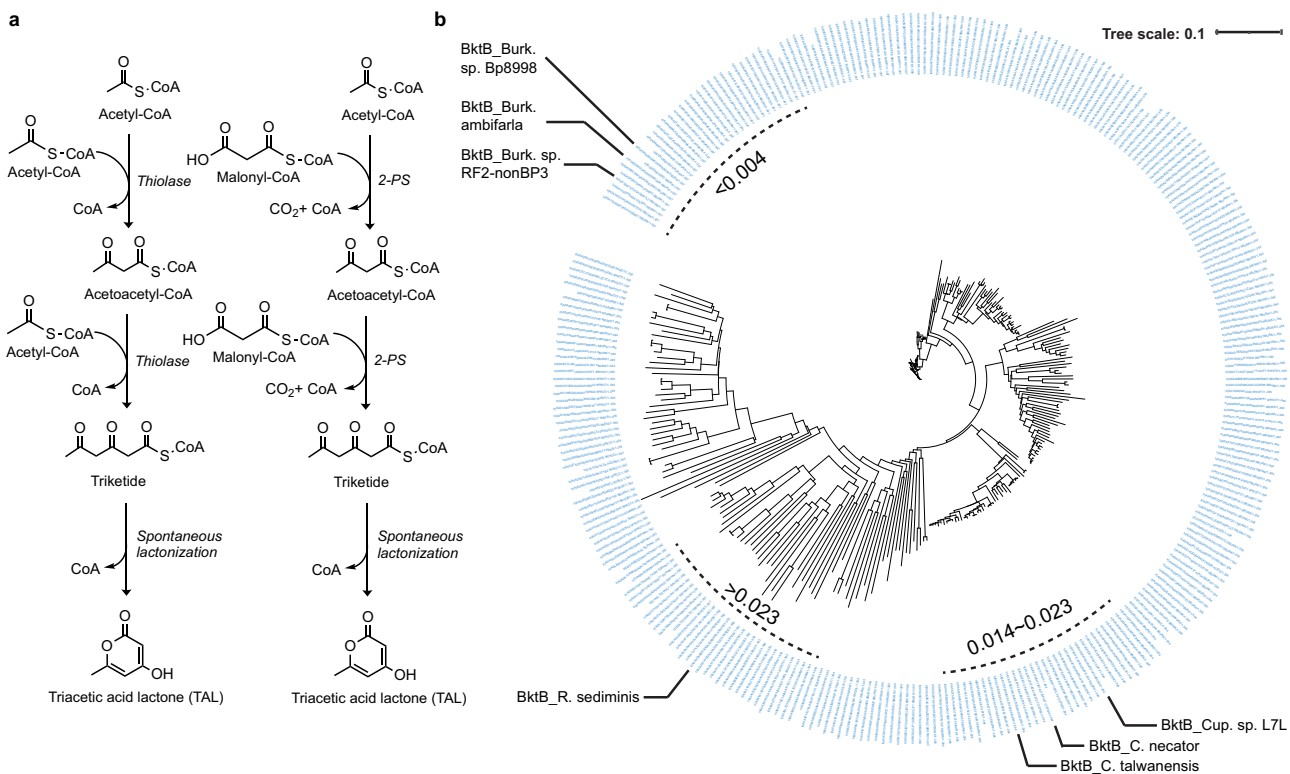

**Fig. 1 | Biosynthetic pathways for triacetic acid lactone and evolutionary analysis of BktBcn homologs. a** Two pathways for TAL production catalyzed by BktB (left) and 2-PS (right). **b** Circular phylogenetic tree constructed with 1798 BktB sequences homologous to the α-helix-3 and α-helix-5 of BktBcn obtained from MAFFT alignment. Other regions of proteins were excluded from the alignment. Dashed arcs indicate average pairwise evolutionary distances. BktBba is the BktB from *Burkholderia ambifaria*; BktBbr, *Burkholderia* sp. RF2-non_BP3; BktBbb, *Burkholderia* sp. Bp8998; BktBct, *Cupriavidus taiwanensis*; BktBcl, *Cupriavidus* sp. L7L; BktBcn, *Cupriavidus necator* BktB and BktBrs, *Rhodosalinus sediminis*.

ketide synthase, 2-pyrone synthase (2-PS) (Fig. 1a)[16]. Heterologous expression of 2-PS enabled TAL synthesis from different carbon sources in several platform microbial hosts like *Escherichia coli*[17–19], *Saccharomyces cerevisiae*[17,20–22], *Yarrowia lipolytica*[14,23,24] and *Rhodotorula toruloides*[25], with some publications reporting titers of more than 20 g L⁻¹. However, use of 2-PS and malonyl-CoA has several disadvantages: ATP consumption to produce malonyl-CoA from acetyl-CoA by acetyl-CoA carboxylase (ACC), allosteric inhibition of malonyl-CoA on ACC, and competition for malonyl-CoA with native pathways like fatty acid biosynthesis[26]. In 2020, Tan et al. discovered that BktB from *Cupriavidus necator* is a polyketoacyl-CoA thiolase and demonstrated its activity to produce polyketides, including TAL, through repetitive non-decarboxylative Claisen condensations with acetyl-CoA as both precursor and extension unit in vitro and in vivo in *E. coli* (Fig. 1a)[27]. The omission of malonyl-CoA by polyketoacyl-CoA thiolase circumvents the disadvantages of 2-PS, so polyketoacyl-CoA thiolase is a promising alternative enzyme for microbial biosynthesis of TAL.

Here, we show a BktB homolog from *Burkholderia* sp. RF2-non_BP3 (BktBbr, WP_059700748.1), discovered through tests on candidates selected from thousands of BktB homologs with different phylogenetic distances to *Cupriavidus necator* BktB (BktBcn). In addition to the advantages of being a polyketoacyl-CoA thiolase described above, its TAL-producing activity in vitro and resultant TAL production in vivo when expressed in *E. coli* are both ~ 30 times higher than those of BktBcn. Scaled-up production in a fed-batch bioreactor leads to TAL titers in excess of 2.8 g L⁻¹. Additionally, we resolve the high-resolution X-ray structure of BktBbr with three different substrates: acetyl-CoA, acetoacetyl-CoA, and butyryl-CoA. The rational mutagenesis based on the structure analysis allows us to further enhance the TAL production by targeting important amino acids in the CoA binding channel.

## Results

### Identification of BktB homologs with activity for triacetic acid lactone production

Unlike most other thiolases, BktBcn is a polyketoacyl-CoA thiolase able to accept acetoacetyl-CoA as the substrate for its non-decarboxylative Claisen condensation, generating TAL through spontaneous cyclization of the triketide CoA. Hereafter, for convenience, all thiolase homologs of BktBcn will be termed "BktB"s. We used MAFTT to align the protein sequence of BktBcn with 2910 thiolase sequences obtained from the Uniprot database. From these sequences, a circular version of a phylogenetic tree with the top 300 sequences was constructed using EMBL-EBI Phylogeny (Fig. 1b). Six BktB candidates from *Burkholderia ambifaria* (BktBba), *Burkholderia* sp. RF2-non_BP3 (BktBbr), *Burkholderia* sp. Bp8998 (BktBbb), *Cupriavidus taiwanensis* (BktBct), *Cupriavidus* sp. L7L (BktBcl), and *Rhodosalinus sediminis* (BktBrs) (Supplementary Table 1, Supplementary Fig. 1) were then selected based on the evolutionary distance in three clusters. Admittedly, there are still many homologs that were not analyzed in this study, but it does not influence our aim of selecting different homologs at distinct evolutionary distances.

To validate that the selected BktB candidates could produce TAL using acetoacetyl-CoA, all seven BktB proteins including BktBcn were expressed in *E. coli* BL21(DE3) and purified for in vitro studies (Supplementary Fig. 2a, 2b). Except for the nearly insoluble BktBrs, which failed to purify (Supplementary Fig. 2a, 2b), all other BktB candidates converted acetyl-CoA and acetoacetyl-CoA into TAL (Fig. 2b, Supplementary Fig. 2c), leading to a hypothesis that TAL-producing activity exists quite broadly in thiolases. LC-MS analysis revealed that BktBs from *Burkholderia* species produced TAL at a faster rate than BktBs from other microorganisms (Supplementary Fig. 2d).

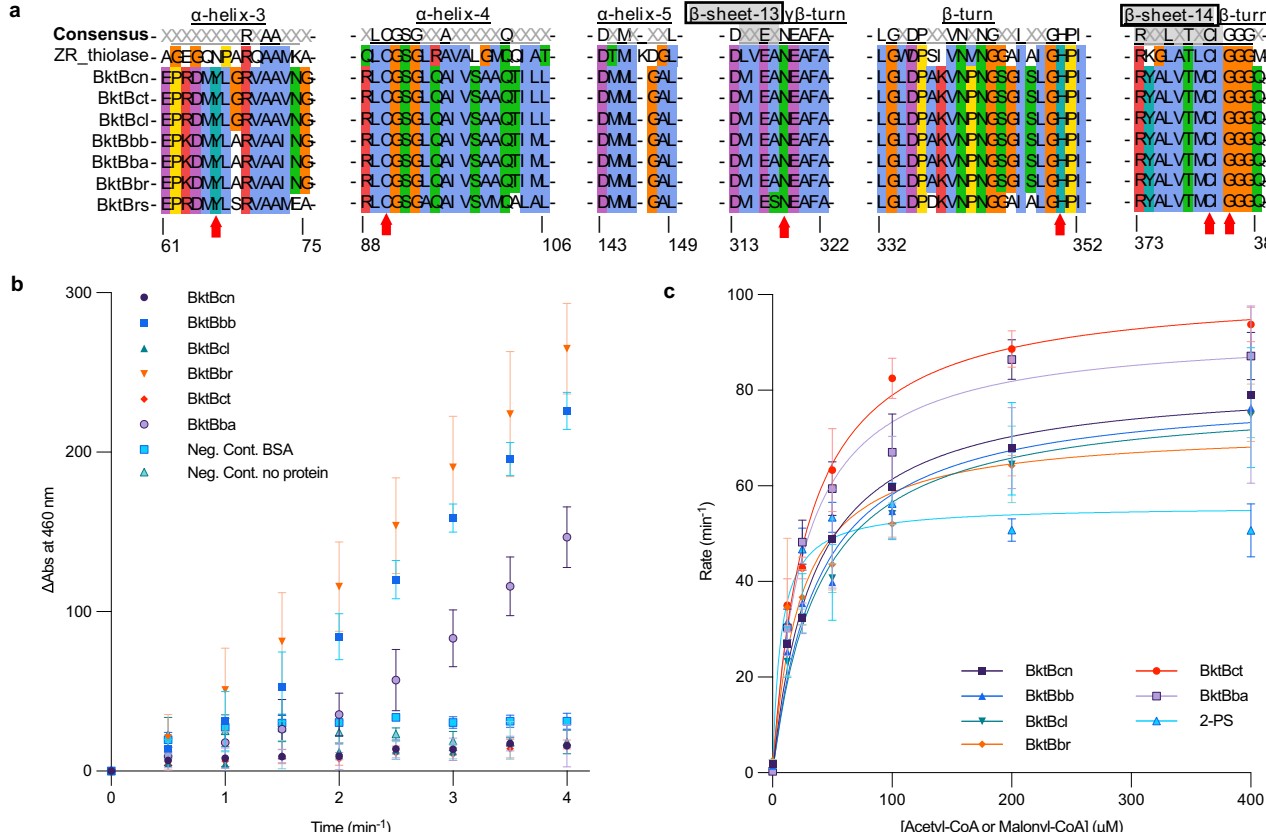

**Fig. 2 | Protein sequence alignment of selected BktB proteins and the kinetics under different conditions. a** Regions of importance including the featured Y66 and α-helix-5, active sites of the sequences are highlighted. PhbAzr is included to show the difference between classic Thiolase II and BktBs. Red arrows indicate amino acids in the active site. X represents the non-consensus amino acids in the alignment. The α-helices, β-sheets and β-turns are underlined, and the β-sheet is boxed in black in the consensus row. Amino acid positions are referenced to those in BktBcn. **b** Enzyme assays of six soluble BktBs with both acetyl-CoA and acetoacetyl-CoA. The reaction contained 0.2 mM acetyl-CoA and 0.2 mM acetoacetyl-CoA, and the reaction was monitored using a coupled reaction with α-KGDH. The rates of the reactions are indicated in the legend. BSA was used in the negative control. **c** Michaelis-Menten plot of all BktBs and 2-PS with only acetyl-CoA or malonyl-CoA. The reaction contains 12.5, 25, 50, 100, 200, and 400 μM acetyl-CoA or malonyl-CoA and was monitored using the same coupled reaction. The data are from three biological replicates (error bars indicate mean values + standard deviation) for **b**, **c**. Source data are provided as a Source Data file.

## *Burkholderia* BktBs have superior enzyme activity

To better understand how the seven selected BktBs are different from each other, protein sequences and enzyme kinetics were analyzed. With the known structures of thiolase PhbAzr (PDB IDs: 1DLV and 1DM3) and BktBcn (PDB ID: 4W61), accurate secondary structures of the entire amino acid sequence can be depicted. As reported, Y66 is a unique feature of BktBs[28] compared with thiolase PhbAzr, which shares with BktBcn the ability to condense two units of acetyl-CoA into acetoacetyl-CoA[29] but cannot synthesize TAL[30]. In the alignment of the seven BktBs and PhbAzr, we found that these proteins have identical active site residues C90, N317, H350, C380 and G382 (amino acid position number in BktBcn) and also identical α-helix-5, which are important for the thiolase activity[28]. PhbAzr differs from BktBcn by having a shorter α-helix-3 (amino acid sequence PARQAAMK), and the loop region of the β-turn close to active site residue H350 is also different. The *Burkholderia* BktBs we selected to experimentally analyze are nearly identical to BktBcn in the highlighted regions, except for A68, which is only consistent in *Burkholderia* BktBs but not in other BktBs (Fig. 2a).

There are two common ways to measure enzyme activity, either monitoring the decrease of substrates or the increase of products. TAL has an absorbance maximum at 298 nm, and thus this wavelength has been widely used for measuring the kinetics of TAL-producing enzymes, including 2-PS and BktB[18,27]. However, direct monitoring of

TAL absorbance at 298 nm, which overlaps with protein absorbance at 280 nm, is not sensitive. In these studies, we used a coupled assay with α-ketoglutarate dehydrogenase (α-KGDH) to produce NADH as a function of the free CoA released during TAL formation and monitored excitation and emission of NADH at 360 nm and 460 nm, respectively, which is more sensitive and accurate than measuring TAL absorbance. The activities of all six soluble BktBs were compared by reacting with 0.2 mM acetyl-CoA and 0.2 mM acetoacetyl-CoA. All three *Burkholderia* BktBs exhibited much higher activity than all other BktBs, especially BktBbr, which was surprisingly nearly 30-fold higher than BktBcn (Fig. 2b). However, the $k_{cat}$ and $K_M$ were difficult to obtain when the reactions contained various concentrations of the acetoacetyl-CoA and a fixed amount of acetyl-CoA. The TAL synthesis activities of BktBcn, BktBcl, and BktBct were significantly suppressed at high concentrations of acetoacetyl-CoA (200 μM), while those of BktBbb, BktBbr, and BktBba were not (Supplementary Fig. 3a). At low to medium concentrations of acetoacetyl-CoA (12.5-50 μM), the enzyme activity of BktBs was not or was partially suppressed. This phenomenon is likely attributed to the reverse reaction of BktB, wherein it produces two acetyl-CoAs from one acetoacetyl-CoA and one free CoA formed by hydrolysis of acetoacetyl-CoA or acetyl-CoA, suggesting that BktBcn, BktBcl, and BktBct have a propensity to degrade acetoacetyl-CoA when it is present at high concentrations, instead of catalyzing its conversion into TAL. Conversely, BktBs derived from

*Burkholderia* were unaffected by adding even high concentrations of acetoacetyl-CoA. Further analysis showed that BktBcn had a lag phase of 5 min at 200 µM acetoacetyl-CoA before TAL production began but not at lower acetoacetyl-CoA concentrations, whereas BktBbr immediately produced TAL and was unaffected by the acetoacetyl-CoA concentration. To exclude the possibility that the lag was caused by the coupled α-KGDH assay, the concentration of the limiting component thiamine pyrophosphate (TPP), a cofactor vital for energy production and metabolic regulation, was increased to enhance the coupled assay. Additionally, acetyl-CoA was further increased to 2 mM to be tenfold higher than the amount of acetoacetyl-CoA (0.2 mM) to enhance the conversion to TAL (Supplementary Fig. 3b). Both findings indicate that the delay in the conversion to TAL was solely influenced by the enzyme itself and not by any other external conditions or the other components in the assay. We used the synthesis rate of TAL ($k_1$) and thiolysis rate ($k_2$) to explain the lag phase in BktBcn. *Burkholderia* BktBs exhibited elevated TAL synthesis rates (high $k_1$), ensuring continuous conversion of acetoacetyl-CoA to TAL. In other BktBs with less robust TAL synthesis rates (low $k_1$), the lag may occur when thiolysis rate at high acetoacetyl-CoA concentrations temporarily surpasses the TAL synthesis rate, requiring a balance shift favoring TAL synthesis (Supplementary Fig. 3c).

In 2020 when BktBcn was reported, Tan et al. proposed that TAL formation from acetyl-CoA via non-decarboxylative condensation is thermodynamically unfavorable ($\Delta_r G'^o$ = 3.7 kcal mol$^{-1}$)[27] However, we calculated a Gibbs free energy change under physiologically relevant standard conditions of -5.2 kcal mol$^{-1}$, indicating overall favorability (Supplementary Fig. 3d). As expected, all BktBs showed quite comparable TAL production activity with only acetyl-CoA in the reaction. BktBbr and BktBba have slightly higher $k_{cat}/K_M$ than BktBcn and BktBcl (Fig. 2c, Supplementary Table 2). Compared with the mixed substrates of both acetyl-CoA and acetoacetyl-CoA, the limited amount of acetoacetyl-CoA synthesized by BktB itself is the rate-limiting factor on TAL formation by *Burkholderia* BktBs. In contrast, BktBs have 1.5-to-2-fold higher $k_{cat}$ than 2-PS but also 4-to-6-fold higher $K_M$ for acetyl-CoA than 2-PS (Fig. 2c). To further support the conclusion, we have independently quantified TAL formation using LC-MS. This time-course assay was performed by using a fixed, saturating 400 µM concentration of acetoacetyl-CoA and various concentrations of acetyl-CoA for BktBcn and BktBbr (Supplementary Fig. 4). This strategy eliminates confounding effects from the reversible condensation of acetyl-CoA to acetoacetyl-CoA, enabling direct assessment of the productive condensation step. The results confirmed the kinetic trends initially observed and support the activity differences among BktB homologs. Considering the intracellular concentration of acetyl-CoA is generally higher than malonyl-CoA in *E. coli*, BktB could be better for TAL production compared with 2-PS. As such, *Burkholderia* BktBs are potentially very efficient for TAL production in vivo. All of the data presented above demonstrate that *Burkholderia* BktBs are much more efficient TAL producers than 2-PS when sufficient acetoacetyl-CoA and acetyl-CoA are present, providing an alternative enzyme for TAL production in microbes.

## Production of TAL using *E. coli* engineered with BktBs

Genes encoding the tested BktBs along with 2-PS and the negative RFP control were then cloned into pBbA5a[31], a medium-copy BioBrick vector with medium strength promoter, and the resulting vectors were transformed into *E. coli* JBEI-3695 (BW25113 Δ*adhE* Δ*ldhA* Δ*frdBC* Δ*pta*). The *E. coli* strains harboring these resultant plasmids were grown, and the TAL concentration was measured. JBEI-3695 was used as the host strain because its deficiency in most of mixed-acid fermentation enzymes improved the supply of substrate acetyl-CoA, which is converted to another substrate acetoacetyl-CoA by native acetyl-CoA acetyltransferase, AtoB, or by the overexpressed BktBs themselves (Fig. 3a). After 48-hour post-induction growth in 24-well plates at

different temperatures (25, 28, 30 and 37 °C) with glycerol as carbon source, we determined that 25 °C growth led to higher TAL titers than titers at 28, 30, and 37 °C (Supplementary Fig. 5). Strains expressing *Burkholderia* BktBs produced more than 75 mg L$^{-1}$ TAL, while most other BktBs only produced approximately 10 mg L$^{-1}$. The strain overexpressing 2-PS produced approximately 30 mg L$^{-1}$, and strains with RFP and BktBrs produced no detectable TAL. Among three *Burkholderia* BktBs, *E. coli* harboring BktBbr produced 0.3 g L$^{-1}$ TAL, 30 times the titer from BktBcn, resembling their in vitro test results (Fig. 3b). BioBrick vectors with different copy numbers and promoter strengths were then tested for expressing BktBbr, and as expected, the TAL titer improved with vector copy number. Further implementation of a stronger promoter on pBbE1a resulted in 0.8 g L$^{-1}$ TAL (Fig. 3c). JBEI-3695 pBbE1a-bktBbr, the strain with highest TAL production in 24-well plates, was also grown in shake flasks with either glycerol or glucose as carbon sources, resulting in 0.4 g L$^{-1}$ and 0.2 g L$^{-1}$ of TAL, respectively (Fig. 3d).

Fed-batch production of TAL using *E. coli* expressing 2-PS has been well studied[17,18,25]. However, fed-batch production using *E. coli* expressing BktB has been rarely documented. Thus, we performed a 1-L fed-batch production study of *E. coli* JBEI-3695 pBbA5a-bktBbr with glucose or glycerol as a carbon source. During the five-day production, cells grew faster with glycerol, while TAL production was higher with glucose. Compared with ~1.6 g L$^{-1}$ TAL production using glycerol as a carbon source, ~2.8 g L$^{-1}$ TAL was produced using glucose as a carbon source (Fig. 4a, b, Supplementary Fig. 6a, 6b). The TAL yield on glucose was 0.11 g g$^{-1}$. Further analysis of the glucose and secondary metabolite levels during the production indicates that less TAL was produced after 74 h than before 74 h. Although *E. coli* JBEI-3695 had nearly all mixed-acid production enzymes knocked out, the acetic acid was probably produced by the remaining PoxB[32], and then used as a carbon source for growth and TAL production after glucose was exhausted (Supplementary Fig. 6a).

It is commonly accepted that TAL poses some toxicity to *E. coli*. Overcoming TAL toxicity is critical for TAL production in *E. coli*, particularly if a very efficient TAL production enzyme, like BktB, is used. We tested TAL toxicity on two strains, JBEI-3695 and JBEI-3695 pBbE1a-BktBbr (TAL producer), in LB and LB-plus media (richer supplements supporting cell growth and used in TAL production). After monitoring cell growth for 60 h at 37°C, both strains exhibited comparable TAL toxicity in LB (Fig. 5a–b) and LB-plus (Fig. 5c–d). Cells harboring a plasmid expressing BktB experienced higher growth inhibition due to potential production of TAL, presented by worse cell growth with 1 g L$^{-1}$ TAL in LB and 3 g L$^{-1}$ TAL in LB-plus. *E. coli*'s tolerance to TAL was much better in the LB-plus medium with richer supplements than in standard LB. Overall, in LB medium, the cell growth was mildly inhibited below 0.75 g L$^{-1}$ TAL, while a significant inhibition occurred at 1.0 g L$^{-1}$ TAL, and no cell growth observed above 1.5 g L$^{-1}$ TAL. Similarly, in LB-plus medium, the cell growth was mildly inhibited below 2.5 g L$^{-1}$ TAL, while a significant inhibition occurred at 3.5 g L$^{-1}$ TAL, and no cell growth observed above 4.0 g L$^{-1}$ TAL. This could also explain why the highest titer achieved in bioreactors with LB-plus medium was not greater than 2.8 g L$^{-1}$. In addition, we measured the half maximal inhibitory concentration (IC50), representing the potency of TAL in inhibiting the *E. coli* cell growth (Fig. 5e–h). Few studies have characterized the IC50 of TAL in this system and show consistent conclusions with the cell growth assay: in LB medium the IC50 of TAL was ~1.0 g L$^{-1}$, whereas the IC50 in LB-plus medium was ~3.0 g L$^{-1}$. This 3-fold difference indicates that TAL toxicity can be overcome with richer supplements that support cell growth. This finding may lead to a solution to reduce the TAL toxicity for *E. coli* because the richer medium enhances most protein expression levels[33], possibly including transporters responsible for exporting intracellular TAL from the cell. Identification and up-regulation of the TAL transporters in *E. coli* may be important for it to achieve the high TAL titers. Such transporter engineering, for

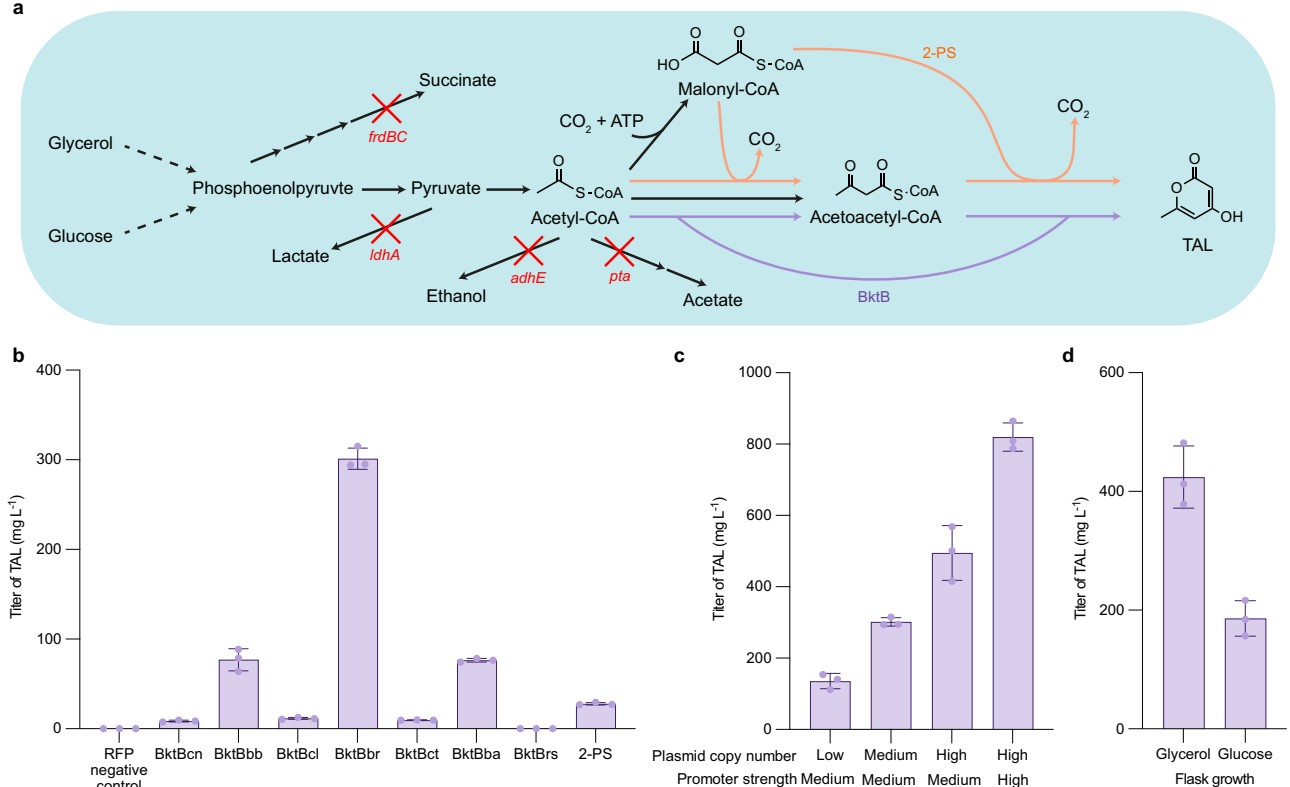

**Fig. 3 | In vivo TAL production. a** Pathway scheme of TAL production in JBEI-3695, an *E. coli* BW25113 strain with most mixed acid fermentation pathways eliminated through knock-out of *adhE*, *ldhA*, *frdBC* and *pta* (indicated by red crosses) to improve the acetyl-CoA supply. Black arrows indicate native pathways related to conversion of carbon sources to TAL; purple arrows indicate the reactions catalyzed by overexpressed BktBs; orange arrows indicate the reactions catalyzed by overexpressed 2-PS. **b** TAL titers of JBEI-3695 expressing the genes encoding different BktBs and 2-PS on pBbA5a, a medium copy number vector with medium strength promoter. **c** TAL titers of JBEI-3695 expressing *bktBbr* on plasmids with different copy numbers and different promoter strengths. Production runs **b**, **c** were performed in 24-well plates with glycerol as carbon source. **d** TAL titers from shake flask cultures of JBEI-3695 pBbE1a-bktBbr, which expresses BktBbr on a high copy number vector with a strong promoter, with glycerol or glucose carbon source. All growths were conducted at 25 °C. The data are from three biological replicates (error bars indicate mean values + standard deviation) for **b**–**d**. Source data are provided as a Source Data file.

example efflux pumps, in *E. coli* has been reported to reduce the toxicity of biofuels[34]. Therefore, further increases of TAL production by *E. coli* may require strain engineering or adaptive laboratory evolution to mitigate TAL toxicity[35]. Additionally, an efficient organic overlay could be also helpful by transferring TAL from the fermentation broth to the overlay.

### Engineering BktBs using high-resolution X-ray structures

There has been no available high-resolution structure of BktB bound with CoA ester substrates, especially with acetoacetyl-CoA which supports the TAL biosynthesis[27]. While an *apo* structure of BktBcn was reported in 2015 at a resolution of 2.0 Å[28], using this structure to investigate the protein-ligand interactions of BktB and the involvement of water molecules in these interactions, essential for thiolases, remains challenging[36]. To enhance TAL production by *Burkholderia* BktBs, a high-resolution structure of BktB is crucial. Here, we successfully obtained X-ray structures of the highly active BktBbr in its *apo* form (PDB ID: 9BWK) and bound with acetyl-CoA, acetoacetyl-CoA, and butyryl-CoA (PDB IDs: 9BWO, 9BWP, 9BWL) at 2.2 Å, 2.2 Å, 2.4 Å and 2.1 Å resolution, respectively (Fig. 6a–d, Supplementary Fig. 7, Supplementary Table 3). While most thiolases exist as dimers[37], BktB exists as a tetramer (BktBcn, PDB ID: 4W61)[28]. BktBbr has a special loop region that stabilizes the tetrameric structure (Supplementary Fig. 8a–b). Deleting this loop region eliminated >95% TAL production (Supplementary Fig. 8c–d). Comparing the *apo* structure with the structures covalently bound by CoA esters, no obvious structural changes were observed. Both acetyl-CoA and butyryl-CoA, acting as starter substrates for TAL and ketoacyl-CoA biosynthesis, respectively, displayed identical protein-ligand interactions: H-bonding between the keto group of substrates with C95 and G387 within the active pocket as well as interactions between the CoA with R226 and S254 (Fig. 6b, d). These findings indicate that the BktBbr active pocket accommodates acyl-CoAs with longer hydrocarbon tails, facilitating condensation within the pocket. Interestingly, interactions crucial for anchoring CoA substrates were not evident in the acetoacetyl-CoA structure. Steric hindrance from the additional keto group in the acetoacetyl group likely eliminated interactions within the active site pocket (Fig. 6c). Instead, substantial interactions between BktB and CoA were identified, mediated by water molecules (Fig. 6c). Water molecules, capable of acting as both donors and acceptors, are pivotal in mediating H-bonds[38,39].

Based on these discoveries, we designed twelve mutants to scrutinize individual amino acids' involvement in protein-ligand interactions (Fig. 6e). We screened mutants in *E. coli* BL21(DE3) cultures in 24-well plates and examined TAL production and BktBbr levels. While *E. coli* BL21(DE3) is not the best strain for TAL production due to low intracellular levels of acetyl-CoA and acetoacetyl-CoA compared with the specifically engineered JBEI-3695 strain, BktB expression can be enhanced in *E. coli* BL21(DE3) for both wild-type protein and mutants, significantly reducing the impact from the difference of the protein expression level for comparison (Supplementary Fig. 9a). Proteomics data indicated comparable protein expression levels among all

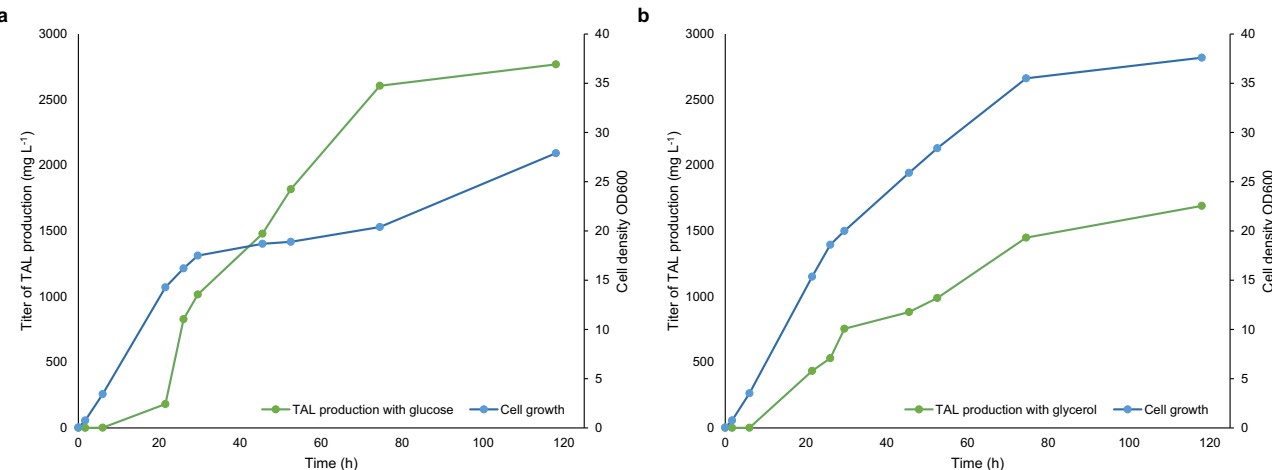

**Fig. 4 | Fed-batch TAL production.** *E. coli* JBEI-3695 with pBbA5a-BktBbr was used for TAL production with **a** glucose or **b** glycerol as carbon source. The production studies were performed over a 5-day period. IPTG was added for protein expression at 4 h. Carbon source feeding was controlled using a dissolved oxygen probe.

Samples for measurement of cell density ($OD_{600}$) and TAL titer taken were removed from the bioreactor at different time points. Source data are provided as a Source Data file.

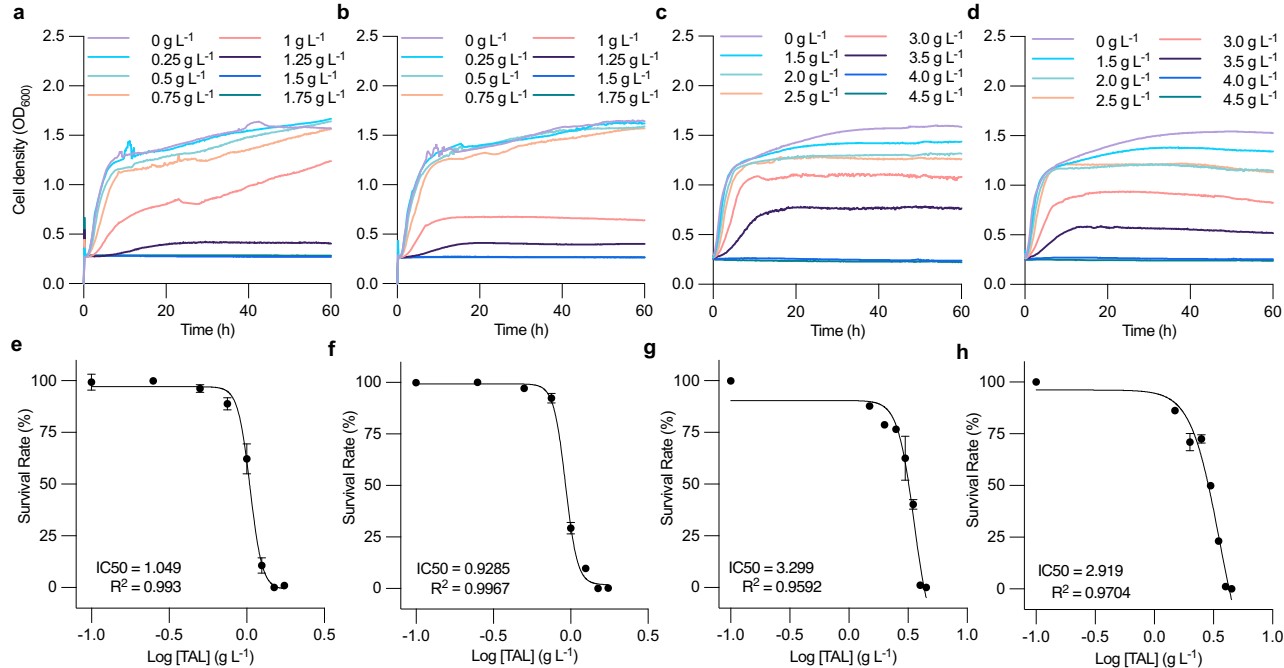

**Fig. 5 | TAL toxicity to *E. coli* in different media.** Cell growth in LB medium: **a** *E. coli* JBEI-3695, **b** *E. coli* JBEI-3695 pBbE1a-BktBbr, and LB-plus medium: **c** *E. coli* JBEI-3695, **d** *E. coli* JBEI-3695 pBbE1a-BktBbr. IC50 results in LB medium **e** *E. coli* JBEI-3695, and **f** *E. coli* JBEI-3695 pBbE1a-BktBbr, and LB-plus medium **g** *E. coli* JBEI-3695

and **h** *E. coli* JBEI-3695 pBbE1a-BktBbr. The incubation was conducted in a 48-well plate at 37 °C over 60 h. Cell growth was monitored by measuring $OD_{600}$. The data are from three biological replicates (error bars indicate mean values + standard deviation). Source data are provided as a Source Data file.

mutants, ranging within ±30% of wild-type BktBbr levels. Consistent with existing literature, C95S in the active site pocket abolished enzyme activity. Intriguingly, the I256A mutation, located within the CoA binding tunnel, led to near-complete loss of enzymatic activity. Structural analysis (Supplementary Fig. 8e–h) revealed that substitution of the bulkier isoleucine with alanine disrupted key hydrophobic interactions that normally stabilize CoA positioning within the tunnel. Specifically, Supplementary Fig. 8e–g show the original narrow tunnel in wild-type BktBbr, while Supplementary Fig. 8h illustrates the expanded tunnel geometry in the I256A mutant. As a result, CoA binding was destabilized, likely impairing proper substrate orientation

for Claisen condensation. These structural changes explain the drastic reduction in catalytic efficiency observed for the I256A variant. Among the mutants, S254A, G255A, and G251A substantially boosted TAL production. A shared trait of these mutants was reduced CoA binding affinity in the substrate tunnel. This aligns with the finding in the recent machine learning-guided mutagenesis of Tfu_0875, a thermostable β-ketothiolase from *Thermobifida fusca*, that the substrate tunnel of thiolases plays an important role in the binding and condensation of substrates, directly affecting enzyme activity[40]. S254A, located adjacent to the CoA tunnel as observed in the substrate-bound BktBbr structure, led to over a two-fold increase in TAL production, while

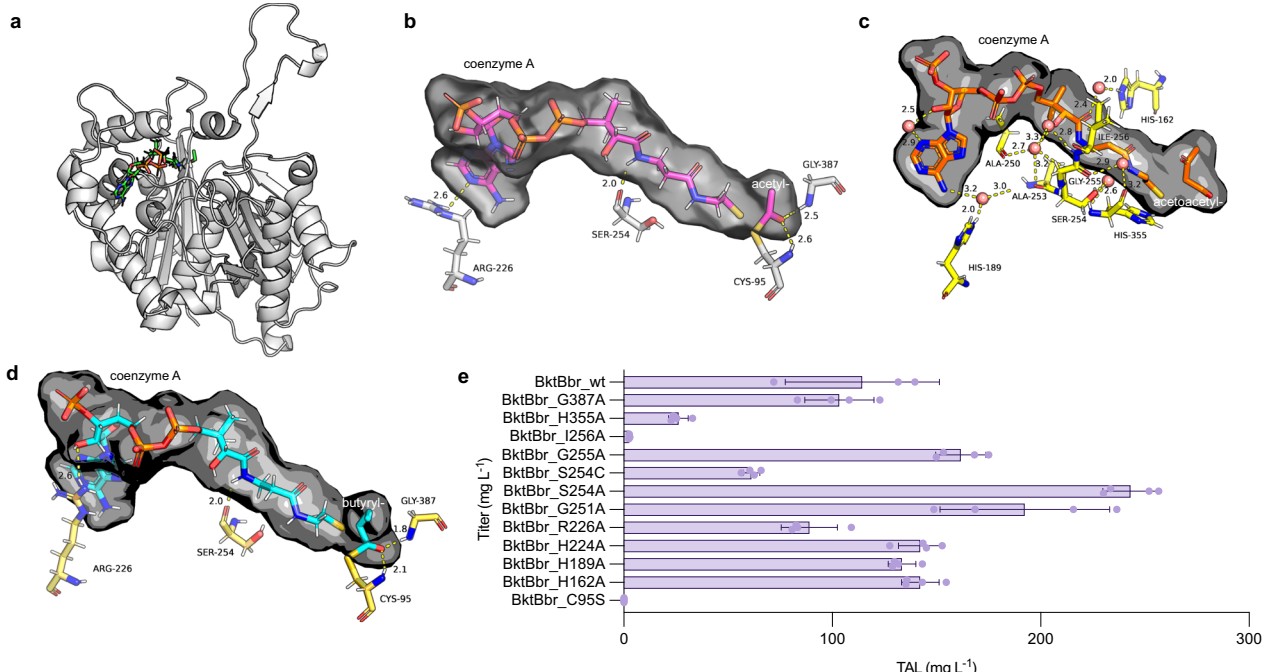

**Fig. 6 | Structure-based engineering of BktBbr. a** A single-chain representation of the structure of BktBbr with acetyl-CoA bound (PDB ID: 9BWO). **b** Binding of acetyl-CoA in the active site (PDB ID: 9BWO). **c** Interaction of acetoacetyl-CoA with BktBbr through water molecules (PDB ID: 9BWP). **d** Binding of butyryl-CoA in the active site (PDB ID: 9BWL). The CoA substrate surfaces are rendered in translucent black, and amino acid residues that interact with CoA, or water molecules are annotated. Water molecules are depicted as red spheres, and polar contacts between these water molecules are in yellow color. Distances of these polar contacts are indicated. **e** TAL production using mutated BktBbr with single amino acid residues interacting with the three different substrates. Microbial production was conducted in a 24-well plate using engineered *E. coli* BL21(DE3) harboring a plasmid with the mutants. All growths were conducted using glycerol as the carbon source at 25 °C for 48 h. The data are from four biological replicates (error bars indicate mean values + standard deviation). Source data are provided as a Source Data file.

S254C decreased it, implying that not just the water network but also CoA chain movement are vital for enzyme activity. Structural comparison suggests that this serine side chain may contribute to spatial constraints at the entrance of the tunnel, and substitution with a smaller alanine could reduce steric hindrance and enhance substrate flux. While we have not directly measured the impact on CoA binding kinetics, this result highlights the potential of tunnel-facing residues as targets for activity enhancement. Furthermore, purified mutants of S254A and G251A exhibited overall better enzyme activity for TAL formation in in vitro assays, while G255A performed comparably to the wild-type (Supplementary Fig. 9b–c). BktBbr_G251A showed ~50% increase in enzyme activity at high concentrations of acetoacetyl-CoA compared to the wild-type BktBbr. This result further demonstrates the consistency between in vivo and in vitro results.

Comparison of the previously reported *apo*-BktBcn structure[28] with BktBbr bound to acetyl-CoA showed a root mean square deviation (R.M.S.D.) between the structures of 0.322 Å, revealing a high degree of structural similarity between BktBcn and BktBbr (Fig. 7a). This similarity underscores that even minor distinctions between these two enzymes can have a profound impact on their enzymatic activities. It is conceivable that these slight discrepancies, particularly in the vicinity of the CoA substrate tunnel and the active site pocket, may be accountable for the substantial differences in enzymatic activity. An examination of the active site pocket reveals an almost indistinguishable configuration of amino acid residues in the pockets of BktBcn and BktBbr (Fig. 7c). To delve deeper, we executed a comprehensive exchange of distinct motifs between the two BktBs, encompassing both proximal and distal regions relative to the active sites and the substrate tunnel. The variants of BktBcn were expressed in *E. coli* BL21(DE3) to ensure elevated protein levels. With the exception of Mutant-7, all other hybrid enzymes produced comparable levels of TAL (Fig. 7b, Supplementary Table 4). Noteworthy is Mutant-7

featuring the AES117-119SEN alteration, which yielded a 2.5-fold increase in TAL production. AES117-119SEN is distant from the active site pocket. Guided by the predictions of AlphaFold 2, we generated a structural model of Mutant-7 and superimposed it onto the wild-type BktBbr structure. Interestingly, this mutant configuration does not change the backbone structure after superposing the X-ray structure with the AlphaFold predicted structure (Fig. 7d). The only differences are the side chains of the amino acid residues between alanine to serine and serine to asparagine, which potentially influence the enzyme's dynamics and activity. Purified BktBcn_AES117-119SEN also showed ~30 % increase in enzyme activity at 50 and 100 μM acetoacetyl-CoA and remained comparable with the wild-type BktBcn at 12.5 and 200 μM acetoacetyl-CoA (Supplementary Fig. 9b–c), implying that the intracellular acetoacetyl-CoA concentration was probably between 50-100 μM, which was favorable by this mutant to produce more TAL. BktBcn_AES117-119SEN also exhibited a similar lag phase as the wild-type BktBcn (Supplementary Fig. 9d), indicating similar kinetics of the mutant compared to the wild-type. These 21 mutants were first guided by sequence divergence between BktBcn and BktBbr, including BktBcn_AES117-119SEN. Following the identification of activity-enhancing mutations, we mapped these residues onto the solved structure, which revealed their proximity to the substrate tunnel or active site. This post hoc structural interpretation offers mechanistic insights into how specific residues like S254 and G251 influence enzyme activity by modulating CoA accessibility and tunnel dynamics.

## Discussion

As a bio-privileged molecule for organic synthesis in industrial manufacturing[15], TAL has attracted a growing interest. *Yarrowia lipolytica* and *Rhodotorula toruloides* have been engineered to produce 36 g L⁻¹ and 28 g L⁻¹ TAL, respectively[14,25]. However, both studies used 2-PS as the TAL producer. 2-PS consumes intracellular malonyl-CoA,

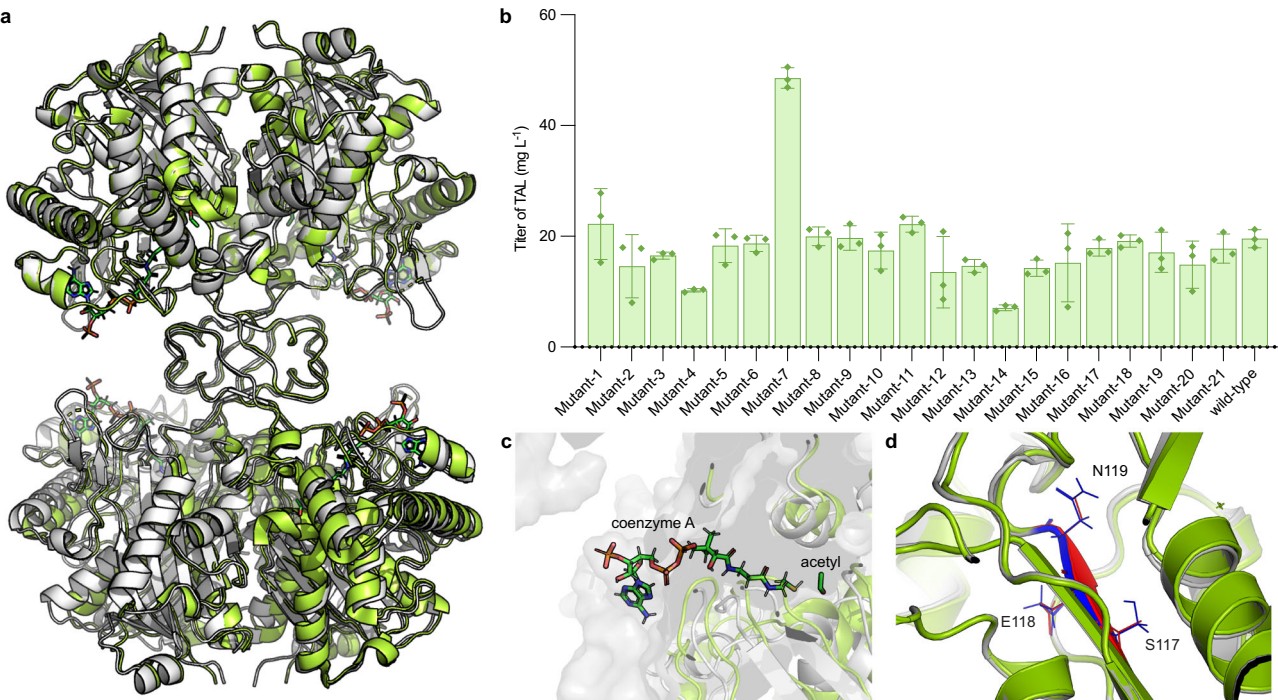

**Fig. 7 | Enhancing the TAL production of BktBcn via the structure comparison between BktBcn and BktBbr. a** Superposition of tetrameric structures of *apo* BktBcn (PDB ID: 4W61) and acetyl-CoA–bound BktBbr (PDB ID: 9BWO). **b** TAL production using mutated BktBcn with swapped comparable motifs from BktBbr. The production was performed in a 24-well plate with the corresponding plasmid transformed into *E. coli* BL21(DE3) with glycerol as the carbon source. All growths were conducted under 25 °C for 48 h. The data are from three biological replicates (error bars indicate mean values + standard deviation). **c** Superimposed binding of acetyl-CoA in *apo* BktBcn. **d** Superposition of X-ray structure of BktBcn with AlphaFold predicted structure of mutant-7 with AES117-119SEN. The RMSD = 0.318. Wild-type BktBcn structure is represented in green, while the predicted structure of BktBcn mutant-7 (from AlphaFold2) is gray. The amino acids structure of AES in wild-type BktBcn is red, and the amino acids structure of SEN in mutant-7 is blue. Amino acid residue structures are represented as lines. Source data are provided as a Source Data file.

which is critical for cell growth and maintaining other basic cell activities, and whose supply from acetyl-CoA suffers from high energy use to fix carbon dioxide and allosteric feedback inhibition. In this scenario, polyketoacyl-CoA thiolase BktB exhibits advantages as an alternative TAL producer by bypassing malonyl-CoA and directly using acetyl-CoA, one of the most abundant intracellular CoA intermediates. In this study, BktBs from *Burkholderia* were shown to be superior to BktB from *C. necator* for TAL production in *E. coli*. Compared with the conserved amino acids found in BktBcn, a phylogenetic analysis enabled us to discover other *Burkholderia* BktBs among the numerous thiolases. When the best of these BktBs, BktBbr, was expressed in *E. coli*, the resulting strain produced 2.8 g L⁻¹ TAL, even if the production may have been inhibited by TAL toxicity. This superior TAL-producing enzyme could be used in yeast with enhanced acetyl-CoA and acetoacetyl-CoA levels and high tolerance to TAL toxicity[17,20–22].

While BktB has not yet matched the high titers reported for 2-PS (e.g., 35.9 g L⁻¹ in *Yarrowia lipolytica*), it offers a mechanistically distinct route for TAL synthesis via acetyl-CoA and non-decarboxylative Claisen condensations. This route avoids ATP consumption and $CO_2$ release, resulting in improved theoretical carbon and energy efficiency. The higher yield observed in BktB-mediated production (0.11 g TAL g⁻¹ glucose) compared to 2-PS–based processes (0.067 g TAL g⁻¹ glucose spiked with acetate) demonstrate this advantage. Our work demonstrates that BktB can be significantly improved through homolog discovery, rational engineering, and production optimization, and serves as a complementary strategy to 2-PS–based systems.

High-resolution X-ray structures of the *apo* enzyme and enzyme bound to various CoA substrates allowed us to perform rational mutagenesis and generate mutants that were more productive when expressed in *E. coli*. The most productive mutants, BktBbr_S254A and

BktBbr_G251A, had correspondingly higher activity in vitro than the wild-type enzyme. By superimposing the structures of BktBcn and BktBr, we were able to identify a three-amino-acid region that, when grafted into BktBcn, improved TAL production by 2.5-fold. Structural comparison revealed several key differences in the substrate tunnel and CoA binding region that correlate with catalytic activity. Residues near the catalytic tunnel entrance, such as S254 and G251 in BktBbr, are positioned to modulate CoA accessibility and potentially stabilize transition states during chain elongation. Our structure-guided mutagenesis, targeting these tunnel-proximal residues, led to significant improvements in TAL production, supporting their functional importance. The tunnel architecture in BktBbr appears optimized to facilitate substrate positioning and product release, which may underlie its superior $k_{cat}$ and in vivo product titer compared to BktBcn. These findings demonstrate that subtle tuning of substrate tunnel geometry and CoA coordination plays a critical role in enhancing polyketide thiolase activity, providing a blueprint for future enzyme engineering efforts.

Given the abundance of secondary metabolites that may be naturally synthesized from TAL[41], it is likely that even more efficient BktB variants remain to be discovered. Continued exploration of BktB homologs, informed by structural and functional insights, will deepen our understanding of thiolase evolution and further advance sustainable biomanufacturing of valuable chemicals.

## Methods
### Strain construction
Strain JBEI-3695 was constructed through step-by-step knockouts of *adhE*, *ldhA*, *frdBC* and *pta* in *E. coli* K-12 BW25113. Each step of knockout was conducted through λ-Red-mediated recombination based

replacement of target gene with kanamycin resistance marker amplified from corresponding strains of Keio Collection, a library of *E. coli* strains with each single gene replaced with kanamycin resistance[42], followed by FLP-mediated removal of resistance[43]. All strains were stored in 30% glycerol under -80 °C. Strains were streaked or plated on LB agarose plates (Teknova, Hollister, CA) with addition of antibiotics at the following concentrations if necessary: carbenicillin (100 μg mL[-1]) and kanamycin (50 μg mL[-1]). Unless specified, all media components and supplemented chemicals for strain growths and constructions were acquired from Sigma-Aldrich (St. Louis, MO), Thermo Fisher Scientific (Waltham, MA) and Becton, Dickinson and Company (Franklin Lakes, NJ).

## Plasmid construction

All genes encoding BktBs and 2-PS were synthesized with GenSmart™ codon optimization, based on the "Population Immune Algorithm", for *E. coli* and subcloned into pET-28a(+)-TEV vector between NdeI/XhoI restriction sites by Genscript (Piscataway, NJ). Genbank Accession Numbers of proteins tested in this study can be seen in Supplementary Table 1, and the list of plasmids used in this study can be seen in Supplementary Table 5. Other plasmids were constructed through Gibson Assembly between vector and genetic parts amplified by PCR with Phusion High-Fidelity DNA polymerase with all kits from New England Biolabs (Ipswich, MA). The PCR amplified DNAs were purified by DNA Clean and Concentrator or Zymoclean™ Gel DNA Recovery Kit (Zymo Research, Irvine, CA). Primers were synthesized by Integrated DNA Technologies (Coralville, IA). Biobrick plasmids[31] with different copy numbers and promoter strengths were used as vectors. The constructed plasmids were chemically transformed to DH5a competent cells (New England Biolabs) and then their sequences were confirmed through Sanger Sequencing by Sequetech Corporation (Mountain View, CA). The sequence confirmed plasmids were then transformed into JBEI-3695 or other host strains through TSS chemical transformation.

## Protein purification

*E. coli* BL21(DE3) transformed with pET-28a (+)-TEV harboring the gene encoding BktBs or 2-PS was inoculated into 10 mL of LB broth containing kanamycin, incubated at 37 °C overnight at 225 rpm shaking. This seed culture was used to inoculate 1 L of LB broth containing kanamycin and then incubated at 37 °C for 2 to 3 hours at 225 rpm shaking. When the OD$_{600}$ reached 0.6, the culture was cooled on ice for 15 - 20 min. 0.2 mM isopropyl β-D-1-thiogalactopyranoside (IPTG) was added to induce protein expression followed by incubation at 18 °C to allow the protein expression. Cells were harvested by centrifugation at 5,000 × g at 4 °C. Harvested cells were transferred to a 50 mL Falcon tube kept at -80 °C until needed. Pellets from 0.2 L of cell culture were thawed at room temperature and resuspended in a 10 mL cell lysis buffer (50 mM sodium phosphate, pH 7.4, 300 mM NaCl, 10% glycerol). Cells were lysed by sonication for 15 min (500-Watt, 30% amplitude, pulse 5 s on and 10 s off). Cell debris and other insoluble material were removed by centrifugation for 45 min at 13,000 × g at 4 °C. The soluble fraction was filtered using a 0.22 μm filter, mixed with 1 mL of Ni-NTA resin (Thermo Scientific, USA), and incubated at 4 °C for 1 hour under constant mixing. The resulting resin was washed twice with 10 mL wash buffer (50 mM sodium phosphate, pH 7.4, 0.3 M NaCl, 20 mM imidazole), and the target protein was eluted using 5 mL elution buffer (50 mM sodium phosphate, pH 7.4, 0.3 M NaCl, 200 mM imidazole). Purity of the target proteins was checked using an SDS-PAGE gel. The protein concentration was quantified by Bradford Assays using the BSA standard curve. Purified proteins were kept at -80 °C in small aliquots until needed.

## Protein sequence alignment and phylogenetic tree construction

Based on the previous findings[28] about *C. necator* BktB structure (PDB ID: 4W61) possessing unique features in α-helix-3 and α-helix-5 that expand the substrate binding pocket relative to PhbAzr, a CoA-bound biosynthetic thiolase from *Z. ramigera* (PDB IDs: 1DLV and 1DM3)[28], an additional analysis focusing on the α-helix-3 and α-helix-5 was performed, which filtered the sequence pool from 2910 to 1798 candidates. Alignment of the protein sequences obtained from Uniprot (https://www.uniprot.org/) were performed in MAFTT (https://mafft.cbrc.jp/alignment/server/) using UniRef50. The aligned sequences were analyzed on ClustalW2 Phylogeny of EMBL-EBI (www.ebi.ac.uk/Tools/phylogeny/) by applying neighbor-joining clustering methods. Visualization of the phylogenetic tree was generated by iTOL (itol.embl.de/tree/). The mode of the tree is circular with rotation of 210 ° and arc of 350°. The branch lengths were used to represent the circular tree.

## In vitro assays

The in vitro assays for functional determination of TAL formation by LC-MS were performed in a 200 μl total reaction volume containing 3 mM EDTA, 1 mM acetyl-CoA and 25 μg (~3 μM) of freshly purified enzymes (noted that *Burkholdeira* BktBs were unstable after freezing-thawing) in 100 mM potassium phosphate buffer, pH 7.2, with or without 1 mM acetoacetyl-CoA. The reaction was incubated at room temperature for 30 min before LC-MS analysis for TAL synthesis. Negative control includes the inactivated BktBs pretreated at 98 °C for 20 min.

The in vitro assays for kinetics using α-KGDH for CoA conversion has been reported[44]. α-KGDH plays a crucial role in the Krebs cycle, facilitating a non-equilibrium reaction where α-ketoglutarate, coenzyme A, and NAD$^+$ are converted to succinyl-CoA, NADH, and CO$_2$. For BktBs, enzyme activities were performed at 22 °C in a 100 μL total reaction volume containing 50 mM sodium phosphate, 3 mM EDTA, 1 mM dithiothreitol, 5 mU μL[-1] α-ketoglutarate dehydrogenase, 2.5 mM NAD$^+$, 0.4 mM thiamine pyrophosphate, 2 mM α-ketoglutaric acid, 200 μM acetoacetyl-CoA, 200 μM acetyl-CoA, and 3 μM of purified enzyme. For the kinetics with only acetyl-CoA (for BktBs) or malonyl-CoA (for 2-PS), the gradients were set to 0, 12.5, 25, 50, 100, 200, 400 μM; for the kinetics with both acetyl-CoA and acetoacetyl-CoA, the gradients of acetoacetyl-CoA were set to 0, 12.5, 25, 50, 100, 200 μM, with the fixed amount of acetyl-CoA at 200 μM. The increase in absorbance at 460 nm (NADH generation) was monitored in a SpectraMax M2 (Molecular Devices, USA) microplate reader for 30 min. Three biological replicates were performed, and the samples were mixed in the 96-well plate with Liquidator™ 96-channel benchtop pipettor (Rainin, USA) to keep the consistency. Data points in the initial 4 min were used to generate the linear curve for the quantification of the initial reaction rate $v$, slope of the linear curve.

For the direct in vitro kinetic assays, we quantified TAL formation over time using LC-MS. The reactions were set up with a fixed, saturating concentration of acetoacetyl-CoA (400 μM) and varying concentrations of acetyl-CoA at 0, 6.25, 12.5, 50, 100, and 400 μM. Samples were collected at 0, 5, 15, 30, and 45 minutes and analyzed by the LC-MS method described below.

## In vivo TAL production

The cell growth medium for TAL production, LB-plus medium, was EZ-rich media (Teknova, Hollister, CA) supplemented with 10 g L[-1] tryptone, 5 g L[-1] yeast extract, 5 mM calcium pantothenate and 20 g L[-1] glycerol or glucose carbon source and with replacement of 1.32 mM K$_2$HPO$_4$ to 2.8 mM Na$_2$HPO$_4$ when glycerol was the carbon source, modified from a previous report[45] by replacing 125 mM MOPS media[46] with EZ-rich media and omitting extra additions of FeSO$_4$, (NH$_4$)$_2$SO$_4$ and NH$_4$Cl. During comparisons of different BktBs, cells were grown in 2.5 mL media contained in Axygen® 24-well Clear V-Bottom 10 mL Polypropylene Rectangular Well Deep Well Plates (Corning, Corning, NY) sealed with sterile AeraSeal™ films (Excel Scientific, Victorville, CA), while the flask growth was performed in 20 mL media contained in 250 mL Pyrex narrow mouth Erlenmeyer flasks with screw cap

(Corning). Before cell growth, a small portion of glycerol stock was inoculated into a glass tube containing 5 mL LB with carbenicillin and grown overnight under 37 °C. 1% volume of overnight seed cultures were then inoculated in the media followed by incubation under 37 °C initially. After three to four hours when cell growth reached exponential phase, 100 µM IPTG was added for induction and cells were transferred to Kuhner ISF-1-W incubators (Kuhner Shaker, Birsfelden, Switzerland) set at 200 rpm shaking and different temperatures of 37, 30, 28 and 25 °C as needed. Cell cultures were collected for analysis after 48 h of post-induction growth. At least three biological replicates were grown in each round, and pre-screening of the colony with highest titer was performed if colony-to-colony variance was high.

### LC-MS TAL identification and quantification

TAL analysis was performed using an Agilent LC/MSD IQ system (Agilent Technologies, Santa Clara, USA). Both in vitro and in vivo, 50 - 100 µL samples were mixed with equal volume of 100% acetonitrile containing internal standard, (5S,6S)-6-isopropyl-5-methyldihydro-2H-pyran-2,4(3H)-dione ($C_9H_{14}O_3$, 170.21 Da), followed by filtration with 3 MWCO 96-well plate (Pall Corporation, Port Washington, NY). Kinetex XB-C18, 2.6 µm, 3 mm×100 mm column (Phenomenex, USA) was used for LC separation of the molecules. A 3 µL sample was injected in the LC-MS for analysis. Mobile phase A was water with 0.1% formic acid, and mobile phase B was methanol with 0.1% formic acid. The detailed method and parameters are shown in Supplementary Table 6. MS was run in negative mode, leading to $m/z$ of TAL is 125.1 and $m/z$ of internal standard is 169.1. Target peaks in SIM mode were auto integrated for peak area. The peak area of TAL is calibrated by the internal standard signal. Three technical replicates were analyzed, with negative controls removing proteins and substrates. All data were analyzed in Agilent OpenLab software.

### HPLC TAL quantification

The TAL titers were quantified using an Agilent 1200 HPLC System (Agilent Technologies, Santa Clara, CA) with Diode Array Detector (DAD) under 298 nm. 500 µL of collected TAL production broth was mixed with 500 µL acetonitrile, followed by 5000 × g centrifuge for 5 min. 300 µL of supernatant was filtered under 3 kDa through Amicon Ultra-0.5 mL Centrifugal Filters (Millipore, Burlington, MA) or 96-well plates (Pall Corporation, Port Washington, NY), then transferred to HPLC vials. Different concentrations of TAL standard samples were also prepared and measured through the same way to create the standard curve. Kinetex 2.6 µm EVO C18 100 Å LC Column with 100 ×4.6 mm size (Phenomenex, Torrance, CA) was used. The sample injection volume was 10 µL. The mobile phase and LC program were the same as LC-MS.

### Fed-batch fermentation conditions

Fed-batch bioreactor experiments were performed in 2-L bench top glass fermenters (Biostat B, Sartorius Stedim, Göttingen, Germany) equipped with two 6-blade Rushton impellers. Cells were grown at 37 °C, 200 rpm for 16 h in LB media. All tanks were batched with 1 L of LB-plus media containing glucose or glycerol as carbon source. The bioreactors were inoculated with seed cultures at a starting OD of 0.05. Temperature, agitation, and air flow were maintained constant at 25 °C, 300 rpm and 0.5 vvm, respectively and pH was controlled to 7.0 using 10% (v v⁻¹) $H_2SO_4$ and 14% (v v⁻¹) $NH_4OH$. Protein expression was induced with 0.1 mM IPTG when OD reached 0.6 and temperature was adjusted to 22 °C after induction. Fed-batch experiments employed a DO signal-triggered feeding loop (ΔDO = 15%, Flow rate = 40 mL h⁻¹, Pump duration = 5 min). Glucose feeding started when the initial amount of glucose was depleted with a feed solution containing 600 g L⁻¹ glucose or glycerol and 50 mg L⁻¹ carbenicillin. 2 ml samples were taken in regular intervals and centrifuged at 15,000 × g for 5 min. For analysis, the supernatant was filtered (0.2 mm) and stored at -20 °C, and the cell pellet was stored at -80 °C.

### TAL toxicity test for variable E. coli strains

A single colony of the cell was inoculated in 5 mL LB broth at 37 °C overnight, 220 rpm. Then the overnight culture was adapted in the 5 mL growth medium at 37 °C overnight, 220 rpm on the next day to obtain the seed culture. The seed culture was inoculated into a 48-well plate, each well containing 0.5 mL LB or LB-plus medium (glycerol carbon source) with different concentrations of TAL (0, 2, 4, 6, and 8 g L⁻¹), and 100 ng µL⁻¹ carbenicillin. Note that TAL solubility is reported to be <8.6 g L⁻¹ in $H_2O$, and 8 g L⁻¹ TAL was directly prepared in LB and TAL production media. Optical density readings were recorded at 600 nm. The starting $OD_{600}$ was 0.2 and the incubation lasted for 100 h at 37 °C, with rapid and constant agitation. The plates were securely sealed using a gas-permeable microplate adhesive film (VWR, USA). The experiment was performed in the BioTek Synergy H1 Plate Reader (USA) and the data was analyzed using Prism Graphpad.

### Crystallization, X-Ray data collection and structure determination

The BktB-apo enzyme was concentrated at 12 mg mL⁻¹ and screened against the crystallization set of solutions: Berkeley Screen[47], MCSG-1 (Anatrace), ShotGun (Molecular Dimensions), PEG/Ion, Index, Crystal Screen, Natrix and PEGRx (Hampton Research). Crystals of BktB-apoenzyme were found in 0.1 M lithium sulfate, 0.1 M sodium chloride, 0.1 M Bis-Tris pH 6.5, 20% PEG 3,350 and 10% hexanediol. Crystals of BktB-apoenzyme were soaking for 3 hours with the ligands butyryl-coA, acetyl-coA and acetoacetyl-coA at 20 mM concentration dissolved in a crystallization mother liquor. The crystals of BktB-apoenzyme and BktB in complex with ligands were placed in a reservoir solution containing 20% (v v⁻¹) glycerol, then flash-cooled in liquid nitrogen. The X-ray data sets for BktB structures were collected at the Berkeley Center for Structural Biology beamline 8.2.2 at the Advanced Light Source at Lawrence Berkeley National Laboratory. The data set was processed using the program Xia2[48]. The crystal structure of BktB-apo enzyme was solved by molecular replacement with the program PHASER[49] using a model generated by ALPHAFOLD[50]. The BktB structures in complex with ligands were solved using the initial model of BktB-apo enzyme. The atomic positions obtained from the molecular replacement were used to initiate refinement using the Phenix suite[51]. Structure refinement was performed using the phenix.refine program[52]. Manual rebuilding was done using COOT[53]. Root-mean-square deviations from ideal geometries for bond lengths, bond angles and dihedral angles were calculated with Phenix[52]. The stereochemical quality of all the final models of BktB-apo enzyme, BktB-butyryl-coA, BktB-acetyl-coA and BktB-acetoacetyl-coA were assessed by the program MOLPROBITY[54]. Summary of crystal parameters, data collection, and refinement statistics can be found in Supplementary Table 3.

### Reporting summary

Further information on research design is available in the Nature Portfolio Reporting Summary linked to this article.

## Data availability

DNA sequences of plasmids and strains used in this study are deposited in the JBEI Public Registry (http://public-registry.jbei.org/). Accession codes are provided in Supplementary Table 5. The protein X-ray structure data are deposited in PDB (https://www.rcsb.org/), with accession codes 9BWK, 9BWL, 9BWO, and 9BWP (Supplementary Table 3). Source data are provided with this paper.

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

## Acknowledgements

This work was supported by the U.S. Department of Energy (DOE) Bioenergy Technologies Office award number 1916-1597, the Joint BioEnergy Institute (https://www.jbei.org), which is supported by the DOE, Office of Science, Office of Biological and Environmental Research under contract DE-AC02-05CH11231, the Philomathia Foundation, and the Nancy P. and Richard K. Robins Family Foundation.

## Author contributions

All authors contributed to the conceptualization of the project. J.D.K., Z.W., S.C. wrote the original manuscript with input from all the other authors. J.D.K., P.D.A., T.S.L. and R.W.H. guided the project. Z.W., W.H., and Y.G. conducted experiments of cloning and protein purification. S.C. conducted the strain engineering and flask fermentation experiments. Z.W., W.H., and Y.G. conducted the enzyme kinetics experiment. J.H.P. and A.D. conducted the experiments of protein crystallization and structure resolving. Z.W. and G.L. conducted the TAL purification and method development. Z.W., W.H, and J.K. conducted the fermentation and sample analysis. All authors contributed to writing the final draft and editing. J.D.K. and P.D.A. acquired funding.

## Competing interests

J.D.K. has a financial interest in Demetrix, Maple Bio, Lygos, Napigen, Berkeley Yeast, Zero Acre Farms, Ansa Biotechnologies, Apertor Pharmaceuticals, and Cyklos Materials. The other authors declare that they don't have competing interests.
