## [Peer Review file · Nature Communications]

A highly active *Burkholderia* polyketoacyl-CoA thiolase for production of triacetic acid lactone

Corresponding Author: Professor Jay Keasling

Version 0:

Reviewer comments:

Reviewer #1

(Remarks to the Author)

This study investigates a novel TAL synthesis pathway via the enzyme BktB, which performs non-decarboxylative Claisen condensation reactions. This enzyme has been previously reported by Tan et al (Nature catalysis).

(1). 2-PS has been reported previously as a very efficient enzyme to produce TAL using malonyl-CoA as extending unit. The study here focused on the identification and characterization of non-decarboxylative Claisen condensation enzyme BktB. The author claimed that BktB is faster than the traditional 2-PS decarboxylative reaction. However, no data is shown. Can the author express 2-PS in *E. coli* and obtain the TAL production kinetics and the k_{cat} value, and make a comparison with BktB?

(2) TAL yield of 36 g/L has been reported in *Yarrowia lipolytica*. The authors argued that 2-PS process depletes intracellular malonyl-CoA, ATP consumption by ACC, and allosteric inhibition of ACC by malonyl-CoA. This argument might be true in *E. coli* due to the very limited amount of malonyl-CoA and the acetyl-CoA carboxylase in *E. coli*. However, this argument doesn't hold true in oleaginous yeast. TAL titer of 4.5 g/L was reported in flask in *Yarrowia*, it clearly shows that the superior activity of 2-PS can drain most of malonyl-CoA towards TAL synthesis in this yeast.

(3) Line 85-89 "bioinformatics analysis to identify six BktB candidates based on the structural alignment of α -helix-3 and α -helix-5 protein domains of BktBcn" should be moved to SI part. This is a routine method and shouldn't occupy the limited text space.

(4). The authors coupled BktB activity with α -ketoglutarate dehydrogenase (α -KGDH) and report the production of NADH. For a crude extract system, there are many other CoA-releasing reactions which may cross-react with KGDH. A more rigorous approach to test the BktB activity is to directly measure TAL with HPLC using UV or RID as detector. The authors argued that proteins (Absorbance at 280 nm) is interfering with the assay of TAL (absorbance at 282 nm). This is not legitimate. As protein can be easily filtered before injecting to HPLC.

(5). To accurately probe the enzyme activity of BktB, C^{13} -isotope labelled acetyl-CoA should be used, and the C^{13} -labelled acetoacetyl-CoA or triketide CoA should be captured by LC-MS. With the current quality of data, it is hard to believe that BktB is performing a non-decarboxylative Claisen condensation reaction.

(6) Gibbs free energy of the BktB step was calculated. Please specify whether the authors used 1 mM physiologically-relevant metabolite concentration or other concentration that is much larger than 1 mM.

(7) A sound toxicity study should have IC_{50} value or inhibition constant. This necessitates the plotting of specific growth rate (1/h) with respect to the dosage of TAL. Fig. 5a and 5b are completely identical, this is totally unacceptable.

(8). To which extent that the X-ray results and the protein mutation work contributes to sustainable TAL cell factory? I think the authors overclaimed their findings. This conclusion can only be made after the authors make a thorough comparison of

BktB and 2-PS in different host.

(9) Lines 88-94: How did you perform the additional analysis on the α -helix-3 and α -helix-5 to filter the sequence pool? And did this filtration process significantly change the top 300 sequences?

(10) Lines 143-166: α -KGDH measures the enzyme activity as a function of the free CoA. But the acetoacetyl-CoA degradation, which is thermodynamically favorable (Extended data Fig. 3d), also consumes the free CoA. I think the lag phase of BktBcn at 200 μ M may be because the free CoA produced in TAL synthesis (k_1) was used for the thiolysis of acetoacetyl-CoA (k_2), resulting in failure to detect CoA at the beginning. Thus, TAL may also be produced at the lag phase, which can be confirmed by measuring the absorbance of TAL. This coupled assay should be double checked.

(11) Lines 362-363: Did the mutant BktBcn_AES117-119SEN still have a lag phase as in Extended data Fig. 3b?

Reviewer #2

(Remarks to the Author)

In this paper, the authors discovered the thiolase that synthesizes triacetic acid lactone (TAL) in high yield from the genome database and achieved high production of TAL (~3 g/L) using recombinant *E. coli* by fed-batch culture. First, the authors searched the genome database for homologs of the thiolase (BktBcn) known to synthesize TALs. These were phylogenetically classified, six representative examples were selected, and their activities were elucidated. The results showed that all of them have TAL synthesizing activity. Detailed kinetic analysis of these enzymes revealed that the enzyme from Burkholderia (BktBbr) was the most active enzyme. The authors proposed that the high activity of the BktBbr can be explained by the low activity to convert acetoacetyl-CoA to acetyl-CoA. Next, they attempted to produce TAL using recombinant *E. coli*. The results showed that the BktBbr-harboring strain produced the highest yield of TAL (0.3 g/L). This production was higher than that of the strain with BktBcn enzyme and higher than that of the strain with the polyketide synthase 2-PS, which also synthesizes TAL. Furthermore, TAL production of 2.8 g/L was achieved by fed-batch culture. The toxicity of TAL to *E. coli* suggested that this yield limit is due to the toxicity of TAL.

The authors next performed an X-ray crystallographic analysis of BktBbr. As a result, they succeeded in determining the structures of four states (apo, butyryl-CoA-bound, acetyl-CoA-bound, and acetoacetyl-CoA-bound forms). Based on these structures, site-directed mutagenesis of the CoA-binding site was performed, and the importance of the residues was confirmed to some extent. In addition, the authors confirmed that some of the amino acid substitutions improved TAL synthesis activity. In addition, the authors have succeeded in increasing activity by modifying the less active BktBcn based on the comparison of BktBbr and BktBcn.

This study reports on the PKS-independent microbial production of TAL. However, the reaction and the concept itself were reported by Tan et al. in 2020 (reference 27) previously. Therefore, the importance of this paper is the discovery of a more active BktB homolog, the implementation of fed-batch culture production using this enzyme, and the elucidation of the structure of the substrate-enzyme complex in BktBbr. Of particular importance are the experiments related to the structure and mechanism of why BktBbr is more productive compared to the other enzymes. The authors have identified a useful BktB homolog, which is interesting, but the function of the enzyme is not analyzed enough. The paper did not adequately explain why the newly discovered BktB is highly active from both a structural and enzymatic point of view. In addition, the authors do not present the data required in a typical structural analysis paper, making it difficult to properly evaluate their hypothesis.

To understand the function of BktBbr more accurately, a detailed analysis of the reaction in which two molecules of acetyl-CoA are formed from acetoacetyl-CoA and CoA is recommended. The authors proposed that the speed of this reaction is important, but have not fully tested how the speed of this reaction varies from enzyme to enzyme. The activity of acetoacetyl-CoA to acetyl-CoA and acetoacetyl-CoA to TAL should be carefully discussed independently. In fact, the prior reaction rate differs among enzymes, what causes the difference in the reaction rate should be discussed based on the structure comparison and examined by the site-directed mutagenesis, etc.

The authors did not include the Fo-Fc map which is generally required when co-crystals are obtained by X-ray crystallography. Without this map, it is impossible to determine the quality of the substrate binding state. In addition, in all structures, the acyl moiety seemed to be transferred to the catalytic Cys but this is not explained enough. Fo-Fc map for this active site should be also provided. Where the acyl moieties are accommodated should be also explained more in detail.

Extended Data Fig. 3. I recommend carrying out some computational simulation to further explain the lag phase of BktBcn although I agree with the hypothesis proposed by the authors.

I preferred to include negative control with the inactivated enzymes in Extended Data Fig. 1c.

Line 106, 107, please provide a. data for this explanation.

Line 171-172, please provide the data related to the yield of TAL from only acetyl-CoA.

Line 170-171, Please add the detailed method for calculation of Gibbs free energy.

BktB-acetyl-CoA has relatively lower l/σ and CC1/2 in the outer shell. Please reconsider the resolution.

Extended Data Fig. 7. e-g is not properly explained. Also, the amino acid labels are sometimes hidden by the model. This should be improved. Also, the hetero atom should be colored differently from carbon atoms (eg for histidine). Fig. 6 has

similar issues. This should be modified appropriately.

Line 177-181. Is there any information on the general concentration of malonyl-CoA and acetyl-CoA in E. coli cells?

Extended data Table 4, "coA" should be "CoA"

Do the authors think, the cyclization of triketide is also catalyzed by the enzyme or occurs spontaneously after the release from the enzyme?

Version 1:

Reviewer comments:

Reviewer #1

(Remarks to the Author)

The authors have addressed most of the concerns. Here is a minor comment.

The Gibbs free energy reported in this work is calculated on the basis of standard Gibbs free energy at 1 mol/L and 298K, 1 atm. However, under physiological conditions, the cellular contents of precursors will never reach 1 mol/L. It is best to emphasize this point and report the Gibbs free energy at 1 mmol/L, rather than 1 mol/L. The calculated standard Gibbs free energy at 1M simply doesn't hold true inside a cell. The real driving force, represented by the Gibbs free energy at physiologically-relevant conditions (1 mM), should be mentioned to avoid any misleading information.

Reviewer #2

(Remarks to the Author)

I am also suspicious about the quality of the fitting in the Extended Data Fig. 4. Data with several more points are recommended. Since there is a log phase for TAL production, please explain which time point was used for Michaelis-Menten kinetics at the figure legend of Extended Data Fig. 4. There is some possibility that the result do not fit well into the Michaelis-Menten if the appropriate time point was selected for velocity analysis.

The authors did not reanalyze the data of BktB acetyl-CoA. Normally, the highest resolution in crystallographic analysis should be set at CC1/2 of 0.5 or better. The resolution for this structure should be around 2.5-2.3 Å?

Extended Data Fig. 9d. Should "time (min⁻¹)" be Should "time (min)"?

Version 2:

Reviewer comments:

Reviewer #1

(Remarks to the Author)

Gibbs free energy change of Bktb step can be easily estimated from eQuilibrator 3.0
<https://equilibrator.weizmann.ac.il/>

Please check the attached file containing the Gibbs free energy change of BktB

Equivalently, search by the KEGG metabolite number, then calculate from eQuilibrator 3.0

Both reactions give the Gibbs free energy change of Bktb at the standard conditions ($\Delta_r G^{\circ}$) and physiologically relevant conditions ($\Delta_r G^{\prime}$) are + 25.0 KJ/mol, which all represents the non-decarboxylative Claisen reaction. Given the misleading information provided by the authors, I recommend a rejection of the current manuscript.

While the authors cited Tan's paper (Nature Catalysis 2020, A polyketoacyl-CoA thiolase-dependent pathway for the synthesis of polyketide backbones), Tan's paper also provided the misleading Gibbs free energy change of Bktb step.

Reviewer #2

(Remarks to the Author)

The authors revised the paper appropriately according to the comments made by the referee.

Reviewer #3

(Remarks to the Author)

I believe that the manuscript is fine.

There is agreement between the concerned reviewer and the authors regarding the condensation step (most easily seen in S3D, step 1), which is around +25 kJ/mol (equivalently, +7 kcal/mol, as written by the authors). While this is a large uphill barrier, and there are contexts in biology where the low concentrations of acetoacetyl-CoA and CoA (relative to acetyl-CoA) cause the reaction to run in the condensation direction, most notably in mammalian liver during ketogenesis. While I'm not sure there is enough information available in literature to reliably calculate the thermodynamics of the eventual hydrolysis and ring closure steps (equilibrator says the uncertainty is too high to provide an answer), these steps are strongly forward driven, which will keep acetoacetyl-CoA low and pull the pathway towards the desired product. The other results of the authors show that the pathway does run net forward in engineered cells, which is really the proof of the overall thermodynamics.

In short, while I appreciate the reviewer's attention to thermodynamics, the overall pathway appears to be thermodynamically forward driven sufficiently to be applied for engineering as the authors describe.

Point-to-Point Response to Reviewer Comments

REVIEWER COMMENTS

Reviewer #1 (Remarks to the Author):

This study investigates a novel TAL synthesis pathway via the enzyme BktB, which performs non-decarboxylative Claisen condensation reactions. This enzyme has been previously reported by Tan et al (Nature catalysis).

(1). 2-PS has been reported previously as a very efficient enzyme to produce TAL using malonyl-CoA as extending unit. The study here focused on the identification and characterization of non-decarboxylative Claisen condensation enzyme BktB. The author claimed that BktB is faster than the traditional 2-PS decarboxylative reaction. However, no data is shown. Can the author express 2-PS in *E. coli* and obtain the TAL production kinetics and the k_{cat} value, and make a comparison with BktB?

Response: We apologize if this point was not clear in the manuscript. We purified 2-PS from *E. coli* for the kinetics study in the manuscript. We did not claim that BktB is faster than 2-PS. In fact, we show in the Fig. 2c that the k_{cat}/K_M of 2-PS is 8.09 while those of the BktBs are 2.0~3.5. BktBs have a higher k_{cat} but also higher K_M . Our work emphasizes that BktB catalysis is mechanistically distinct, operating via non-decarboxylative Claisen condensation with acetyl-CoA, without requiring ATP. This offers a complementary strategy to malonyl-CoA-dependent 2-PS-based TAL synthesis, especially in systems where ATP or malonyl-CoA is limiting.

(2) TAL yield of 36 g/L has been reported in *Yarrowia lipolytica*. The authors argued that 2-PS process depletes intracellular malonyl-CoA, ATP consumption by ACC, and allosteric inhibition of ACC by malonyl-CoA. This argument might be true in *E. coli* due to the very limited amount of malonyl-CoA and the acetyl-CoA carboxylase in *E. coli*. However, this argument doesn't hold true in oleaginous yeast. TAL titer of 4.5 g/L was reported in flask in *Yarrowia*, it clearly shows that the superior activity of 2-PS can drain most of malonyl-CoA towards TAL synthesis in this yeast.

Response: We apologize if this point was not clear in the manuscript. We agree that *Yarrowia lipolytica*—as an oleaginous yeast—has greater capacity for malonyl-CoA biosynthesis than *E. coli*. However, the key limitations of 2-PS–based TAL production are not confined to malonyl-CoA availability alone but include two critical factors: **ATP cost** and **regulatory constraints on acetyl-CoA carboxylase (ACC)**.

1. Energy efficiency and yield comparison:

Although *Y. lipolytica* achieved an impressive titer of 35.9 g/L TAL using 2-PS in a 3-L fed-batch bioreactor (K.A. Markham, et al. *Proc. Natl. Acad. Sci.* 2018, 115(9): 2096-2101), this was accomplished using 18% glucose, 13.7 g/L acetate spikes, and extensive pathway engineering. The calculated yield was **0.067 g TAL/g glucose**—not accounting for acetate as an additional carbon source. In contrast, BktB-mediated TAL production in *E. coli* achieved **0.11 g TAL/g glucose**, representing a significantly higher carbon yield. This advantage stems from BktB’s exclusive use of acetyl-CoA via non-decarboxylative Claisen condensations, which bypasses the ATP-consuming carboxylation step required to generate malonyl-CoA.

2. ACC allosteric inhibition and regulatory burden:

Malonyl-CoA biosynthesis via ACC is subject to tight metabolic regulation. Allosteric inhibition of ACC by malonyl-CoA, free CoAs and fatty acyl-CoAs (Brownsey *et al. Biochem Soc Trans.* 2006, 34(2) 223-227) and competition with fatty acid biosynthesis imposes a major burden on pathway balance and scale-up potential. Even in the engineered *Y. lipolytica* strain used for high-titer 2-PS production, significant metabolic resources were redirected to overexpress ACC and boost acetyl-CoA flux through the pyruvate → acetaldehyde → acetate → acetyl-CoA pathway. These interventions underscore the metabolic overhead required to sustain malonyl-CoA supply.

In contrast, the BktB pathway uses acetyl-CoA directly as the sole precursor, **completely decoupling TAL production from the ACC node** and its regulatory liabilities. This intrinsic simplicity not only improves carbon and energy efficiency but also reduces the metabolic burden and complexity of host engineering.

Taken together, we believe BktB represents a broadly applicable and more efficient platform for TAL production—especially in systems where ATP economy, carbon yield, and regulatory robustness are essential. We have revised the manuscript to clarify these distinctions and contextualize our findings beyond *E. coli*.

(3) Line 85-89 "bioinformatics analysis to identify six BktB candidates based on the structural alignment of α -helix-3 and α -helix-5 protein domains of BktBcn" should be moved to SI part. This is a routine method and shouldn't occupy the limited text space.

Response: Thank you for this comment. We have moved the bioinformatics analysis text to the supplementary information section.

(4). The authors coupled BktB activity with α -ketoglutarate dehydrogenase (α -KGDH) and report the production of NADH. For a crude extract system, there are many other CoA-releasing reactions which may cross-react with KGDH. A more rigorous approach to test the BktB activity is to directly measure TAL with HPLC using UV or RID as detector. The authors argued that proteins (Absorbance at 280 nm) is interfering with the assay of TAL (absorbance at 282 nm). This is not legitimate. As protein can be easily filtered before injecting to HPLC.

Response: We agree with the reviewer's concern regarding the α -KGDH coupled assay, although it has been widely used in CoA-coupled assays. However, this is a compromise because we tried to measure TAL production directly with HPLC DAD at 298 nm after removing the proteins by filtration with 3 kDa cut-off filters. DAD was not sensitive enough to support such measurement of μ M level of TAL formation in a 200- μ L reaction. To address this issue, we independently quantified TAL formation using LC-MS, which is much more sensitive. This time-course assay was performed by fixing 400 μ M acetoacetyl-CoA and varying acetyl-CoA, isolating the productive condensation step. These LC-MS results, now presented as Extended Data Figure 4, confirm the kinetic trends initially observed and support the activity differences among BktB homologs.

(5). To accurately probe the enzyme activity of Bktb, C13-isotope labelled acetyl-CoA should be used, and the C13-labelled acetoacetyl-CoA or triketide CoA should be captured by LC-MS. With the current quality of data, it is hard to believe that Bktb is performing a non-decarboxylative Claisen condensation reaction.

Response: We would like to clarify again that the core mechanism—including the non-decarboxylative Claisen condensation steps, C13-labeled substrate tracing, and thermodynamic characterization—has been extensively demonstrated in the original work (Zaigao Tan et al. *Nature Catalysis* **3**, 593–603 (2020)). We don't see how repeating these studies in our paper would add to what has been done previously.

(6) Gibbs free energy of the Bktb step was calculated. Please specify whether the authors used 1 mM physiologically-relevant metabolite concentration or other concentration that is much larger than 1 mM.<<<<<<

Response: We thank the reviewer for this comment. As seen in Extended Data Fig. 3d, the Gibbs free energy changes of BktB reactions mentioned in this manuscript are Standard Gibbs free energy changes calculated through stoichiometric summation of Standard Gibbs free energy of formation for each compound (Gibbs free energy of formation of 1 mole of each compound), which we estimated using the Group Contribution Method under standard conditions (298.15 K, pH 7.3 and an ionic strength of 0.25). Therefore, the Gibbs free energy changes mentioned in this manuscript are not related to the metabolite concentrations. Rather, these Gibbs free energy changes are Standard Gibbs free energy changes.

(7) A sound toxicity study should have IC50 value or inhibition constant. This necessitates the plotting of specific growth rate (1/h) with respect to the dosage of TAL. Fig. 5a and 5b are completely identical, this is totally unacceptable

Response: We really appreciate the reviewer for pointing out that Fig. 5a and 5b are the same. This error was unfortunately caused by different submitted versions of the manuscript. Furthermore, we have performed a thorough IC50 test of the TAL toxicity to *E. coli* with and without TAL production by BktB. These data are shown in Fig. 5e–h.

(8). To which extent that the X-ray results and the protein mutation work contributes to sustainable TAL cell factory? I think the authors overclaimed their findings. This conclusion can only be made after the authors make a thorough comparison of BktB and 2-PS in different host.

Response: We are sorry if this point was not clear. We did not compare BktB and 2-PS, because they have different mechanisms, and significantly, BtkB saves ATP in the overall reaction mechanism. Please refer to the original paper (Zaigao Tan, et al. *Nature Catalysis* **3**, 593–603 (2020)) for the details. We resolved the structure of the BktBbr with different bound substrates. The rational mutagenesis based on the findings from the structures successfully led to higher TAL production, which demonstrated that we are able to further increase the BktB activity for TAL production.

(9) Lines 88-94: How did you perform the additional analysis on the α -helix-3 and α -helix-5 to filter the sequence pool? And did this filtration process significantly change the top 300 sequences?

Response: As published in the BktBcn structure, α -helix-3 and 5 are not found in thiolases that cannot produce TAL. Adding this filter, the E value of the alignment changed the sequence similarity to be over 50%.

(10) Lines 143-166: α -KGDH measures the enzyme activity as a function of the free CoA. But the acetoacetyl-CoA degradation, which is thermodynamically favorable (Extended data Fig. 3d), also consumes the free CoA. I think the lag phase of BktBcn at 200 μ M may be because the free CoA produced in TAL synthesis (k_1) was used for the thiolysis of acetoacetyl-CoA (k_2), resulting in failure to detect CoA at the beginning. Thus, TAL may also be produced at the lag phase, which can be confirmed by measuring the absorbance of TAL. This coupled assay should be double checked.

Response: We appreciate the reviewer's mechanistic suggestion. This is consistent with our hypothesis. The observed lag phase reflects a transient imbalance where CoA released from TAL synthesis (k_1) is consumed by thiolysis (k_2), leading to an apparent lag in free CoA detection. While TAL may form during this phase, its accumulation may be below detection thresholds. We have revised the manuscript text and caption for Extended Data Fig. 3c to clarify that the lag phase reflects net CoA turnover, not necessarily the absence of TAL formation. The rate v in Extended data Fig. 3c is actually the overall rate of CoA release by BktB, as free CoA is produced in TAL synthesis (k_1) and consumed in acetoacetyl-CoA thiolysis (k_2). Initially, when the thiolysis rate temporarily surpasses the TAL synthesis rate, v is negative, indicating that more free CoA is consumed than released, causing failure of CoA detection by the

coupled α -KGDH assay and hence the lag phase. Therefore, your thought is literally right. Inspired by your comments, we realized that the description in the caption of Extended data Fig. 3c in line 750 that v is the rate of consumption of acetoacetyl-CoA is wrong as both TAL synthesis and acetoacetyl-CoA consume acetoacetyl-CoA. We corrected it as 'overall TAL formation rate' or 'overall free CoA release rate'. Still, other than that error, we believe our description of the hypothesis of the lag phase is not wrong and does not conflict with your thought, as we did not claim there is no TAL synthesis during the lag phase, but "*the lag may occur when the thiolysis rate at high acetoacetyl-CoA concentrations temporarily surpasses the TAL synthesis rate, requiring a balance shift favoring TAL synthesis*". There is a possibility that TAL is not detected during the lag phase, but that could be due to the low TAL synthesis rate subdued by acetoacetyl-CoA thiolysis and insufficient time for accumulation of synthesized TAL to exceed the HPLC sensitivity under that low rate.

(11) Lines 362-363: Did the mutant BktBcn_AES117-119SEN still have a lag phase as in Extended data Fig. 3b?

Response: Yes, the mutant still has the lag phase, and is quite similar with the BktBcn wild type (Extended Data Fig. 9d). The kinetics data points of BktBcn and BktBcn_AES117-119SEN nearly overlap, but the BktBcn_AES117-119SEN is slightly higher, indicating a similar enzyme activity. We believe the degradation of the acetoacetyl-CoA was significantly inhibited (lower k_2), leading to the formation of the TAL detectable in α -KGDH assay under higher concentration of acetoacetyl-CoA.

Reviewer #2 (Remarks to the Author):

In this paper, the authors discovered the thiolase that synthesizes triacetic acid lactone (TAL) in high yield from the genome database and achieved high production of TAL (~3 g/L) using recombinant *E. coli* by fed-batch culture. First, the authors searched the genome database for homologs of the thiolase (BktBcn) known to synthesize TALs. These were phylogenetically classified, six representative examples were selected, and their activities were elucidated. The results showed that all of them have TAL synthesizing activity. Detailed kinetic analysis of these enzymes revealed that the enzyme from Burkholderia (BktBbr) was the most active enzyme. The authors proposed that the high activity of the BktBbr can be explained by the low activity to convert acetoacetyl-CoA to acetyl-CoA. Next, they attempted to produce TAL using recombinant

E. coli. The results showed that the BktBbr-harboring strain produced the highest yield of TAL (0.3 g/L). This production was higher than that of the strain with BktBcn enzyme and higher than that of the strain with the polyketide synthase 2-PS, which also synthesizes TAL. Furthermore, TAL production of 2.8 g/L was achieved by fed-batch culture. The toxicity of TAL to *E. coli* suggested that this yield limit is due to the toxicity of TAL.

The authors next performed an X-ray crystallographic analysis of BktBbr. As a result, they succeeded in determining the structures of four states (apo, butyryl-CoA-bound, acetyl-CoA-bound, and acetoacetyl-CoA-bound forms). Based on these structures, site-directed mutagenesis of the CoA-binding site was performed, and the importance of the residues was confirmed to some extent. In addition, the authors confirmed that some of the amino acid substitutions improved TAL synthesis activity. In addition, the authors have succeeded in increasing activity by modifying the less active BktBcn based on the comparison of BktBbr and BktBcn.

This study reports on the PKS-independent microbial production of TAL. However, the reaction and the concept itself were reported by Tan et al. in 2020 (reference 27) previously. Therefore, the importance of this paper is the discovery of a more active BktB homolog, the implementation of fed-batch culture production using this enzyme, and the elucidation of the structure of the substrate-enzyme complex in BktBbr. Of particular importance are the experiments related to the structure and mechanism of why BktBbr is more productive compared to the other enzymes. The authors have identified a useful BktB homolog, which is interesting, but the function of the enzyme is not analyzed enough. The paper did not adequately explain why the newly discovered BktB is highly active from both a structural and enzymatic point of view. In addition, the authors do not present the data required in a typical structural analysis paper, making it difficult to properly evaluate their hypothesis.

Response: We thank the reviewer for highlighting this matter. We now clarify that several beneficial mutations (e.g., S254A, G251A) reside near the substrate tunnel or CoA binding region, as revealed by post hoc mapping to the BktBbr structure. These positions likely influence tunnel accessibility or product release. Our updated Discussion highlights these correlations and supports them with kinetic enhancements observed in both BktBbr and BktBcn mutants.

To understand the function of BktBbr more accurately, a detailed analysis of the

reaction in which two molecules of acetyl-CoA are formed from acetoacetyl-CoA and CoA is recommended. The authors proposed that the speed of this reaction is important, but have not fully tested how the speed of this reaction varies from enzyme to enzyme. The activity of acetoacetyl-CoA to acetyl-CoA and acetoacetyl-CoA to TAL should be carefully discussed independently. If in fact, the prior reaction rate differs among enzymes, what causes the difference in the reaction rate should be discussed based on the structure comparison and examined by the site-directed mutagenesis, etc.

Response: We thank the reviewer for commenting on this question. We also wanted to test the activity from acetoacetyl-CoA to acetyl-CoA and acetoacetyl-CoA to TAL. However, it is not possible to pause the reaction at the formation of the acetoacetyl-CoA because this is a ping-pong mechanism happening in the same active pocket, especially because acetoacetyl-CoA formation by acetyl-CoA is reversible. No mutagenesis would prevent one reaction from happening and allow the other reaction to occur. Even more, as the binding pocket for acetyl and acetoacetyl moieties are identical, very limited information was provided here for the comparison. Instead, we discussed the difference in the CoA binding, before and after the acetoacetyl-CoA formation, shown in the Fig.6 b-c.

The authors did not include the Fo-Fc map which is generally required when co-crystals are obtained by X-ray crystallography. Without this map, it is impossible to determine the quality of the substrate binding state. In addition, in all structures, the acyl moiety seemed to be transferred to the catalytic Cys but this is not explained enough. Fo-Fc map for this active site should be also provided. Where the acyl moieties are accommodated should be also explained more in detail.

Response: We thank the reviewer for this suggestion. We have now included mFo-DFc omit maps in Extended Data Fig. 7 to support the interpretation of substrate binding. The electron density shown corresponds to mFo-DFc omit maps, in which the ligand (CoA and attached group) was temporarily omitted during map calculation to confirm unbiased density at the binding site. The model was subsequently refined with the ligand built into the observed density, as shown in the figure.

The resulting positive difference density confirms the presence and position of the ligands at the catalytic site, countered at 3.0σ . This visualization highlights the binding of the acyl moieties and their proximity to the catalytic Cys95 residue. As also reported

previously for BktBcn (Kim EJ, et al., *Biochem Biophys Res Commun*, 2014), we observe that the acyl moieties are transferred to the catalytic cysteine in our structures as well. We have updated the figure legend and Methods to clarify this point.

I recommend carrying out some computational simulation to further explain the lag phase of BktBcn although I agree with the hypothesis proposed by the authors.

Response: We thank the reviewer for this suggestion. The computational simulation is only a prediction, which would need to be demonstrated further by experiments. We believe that the kinetic study with LC-MS data of TAL production is sufficient and that a computational simulation will not clarify the mechanism further.

I preferred to include negative control with the inactivated enzymes in Extended Data Fig. 1c.

Response: We added the negative control with the inactivated BktBs, which were pretreated at 98 °C for 20 minutes. The corresponding LC-MS data were included in the Extended Data Fig. 1c.

Line 106, 107, please provide a. data for this explanation.

Response: It is a hypothesis based on our results that the six new soluble BktBs are able to form TAL. In the phylogenetic tree that we provided, there are many more having the same qualifications, based on the sequence alignment to form this phylogenetic tree. Therefore, we do not believe that it is necessary to pick several more BktBs to demonstrate their ability to produce TAL.

Line 171-172, please provide the data related to the yield of TAL from only acetyl-CoA.

Response: We have provided the TAL production from only acetyl-CoA with the LCMS detection, shown in Extended Data Fig. 1d.

Line 170-171, Please add the detailed method for calculation of Gibbs free energy.

Response: This is explained in the caption of Extended Data Fig. 3d. We would like to further stress that they are Standard Gibbs free energy change of reactions calculated by Standard Gibbs free energy of formations of compounds, which were estimated by the group contribution method (Jankowski et al. 2008; Caspi et al. 2016).

BktB-acetyl-CoA has relatively lower $I/\sigma I$ and CC1/2 in the outer shell. Please reconsider the resolution.

Response: We have checked the raw data, and we confirm that BktB-acetyl-CoA has relatively lower $I/\sigma I$ and CC1/2 in the outer shell.

Extended Data Fig. 7. e-g is not properly explained. Also, the amino acid labels are sometimes hidden by the model. This should be improved. Also, the hetero atom should be colored differently from carbon atoms (eg for histidine). Fig. 6 has similar issues. This should be modified appropriately.

Response: We thank the reviewer for this suggestion. We have added the explanations to the Extended Data Fig. 7e–g (now Extended Data Fig. 8e–g) in lines 320-328. We also improved the figures with all the labels shown well and colored hetero atoms differently from carbon atoms.

Line 177-181. Is there any information on the general concentration of malonyl-CoA and acetyl-CoA in *E. coli* cells?

Response: Yes. In *E. coli*, the concentration of acetyl-CoA is typically reported to be 0.05 - 1.5 nmol/mg dry cell weight (corresponding to 20 - 600 μ M), while the malonyl-CoA concentration is considerably lower, ranging from 0.01 - 0.23 nmol/mg dry cell weight (or 4 - 90 μ M) depending on growth conditions and carbon source. As such, acetyl-CoA is usually present at significantly higher concentrations than malonyl-CoA in *E. coli*. (Takamura Y, Nomura G. *J Gen Microbiol.* 1988, 134(8):2249-53; Moteallehi-

Ardakani MH, Asad S, Marashi SA, Moghaddasi A, Zarparvar P. *Mol Biotechnol.* 2023, 65(9):1508-1517.)

Extended dataTable 4 , “coA” should be ”CoA”

Response: Thank you for catching this mistake. We corrected it.

Do the authors think, the cyclization of triketide is also catalyzed by the enzyme or occurs spontaneously after the release from the enzyme?

Response: The cyclization of triketide has been demonstrated to happen spontaneously (see Zaigao Tan, et al. *Nature Catalysis* **3**, 593–603 (2020)).

Point-to-Point Response to Reviewer Comments

REVIEWER COMMENTS

Reviewer #1 (Remarks to the Author):

The authors have addressed most of the concerns. Here is a minor comment.

The Gibbs free energy reported in this work is calculated on the basis of standard Gibbs free energy at 1 mol/L and 298K, 1 atm. However, under physiological conditions, the cellular contents of precursors will never reach 1 mol/L. It is best to emphasize this point and report the Gibbs free energy at 1 mmol/L, rather than 1 mol/L. The calculated standard Gibbs free energy at 1M simply doesn't hold true inside a cell. The real driving force, represented by the Gibbs free energy at physiologically-relevant conditions (1 mM), should be mentioned to avoid any misleading information.

Response: Thank you for the comment. We updated the Gibbs free energy described in Extended Data Fig. 3d and its caption from standard Gibbs free energy ΔG° to physiologically relevant Gibbs free energy ΔG^m in which reactant concentrations are all set as physiologically appropriate 1 mM instead of standard 1 M (Flamholz *et al. Nucleic Acids Research* 2011), and data were re-calculated through equation $\Delta G' = \Delta G^\circ + RT \ln Q$, where Q_r is the reaction quotient under 1 mM reactant concentrations. The sentence '*However, we calculated a standard Gibbs free energy change of -1.1 kcal mol⁻¹, indicating overall favorability (Extended Data Fig. 3d).*' in the main text was hence also changed to '*However, we calculated a physiologically appropriate standard Gibbs free energy change under physiologically relevant standard conditions of -5.21.1 kcal mol⁻¹, indicating overall favorability (Extended Data Fig. 3d).*'

Reviewer #2 (Remarks to the Author):

I am also suspicious about the quality of the fitting in the Extended Data Fig. 4. Data with several more points are recommended. Since there is a log phase for TAL production, please explain which time point was used for Michaelis-Menten kinetics at the figure legend of Extended Data Fig. 4. There is some possibility that the result do not fit well into the Michaelis-Menten if the appropriate time point was selected for velocity analysis.

Response: Thank you for this thoughtful comment. We recognize the importance of both appropriate substrate concentration range and time point selection for robust kinetic fitting.

First, regarding the fitting quality: the acetyl-CoA concentrations used (0, 6.25, 12.5, 50, 100, 400 μ M) span a range that includes values below, near, and well above the observed K_M values (3.7 μ M for BktBcn and 7.8 μ M for BktBbr), capturing both the low-substrate and saturation regions. The 6.25 μ M and 12.5 μ M points effectively bracket the K_M range. While additional points could further refine the curve, concentrations

below 5 μM yield TAL levels near or below the LC-MS detection limit (1–3 μM), reducing data reliability. Given this constraint, the fitted curves ($R^2 > 0.95$) provide accurate and reproducible estimates of kinetic parameters.

Second, regarding time points: we assume the reviewer intended to refer to the “lag phase” rather than the “log phase” of TAL production. For BktBcn, we used data from the 5–45 min range, where TAL accumulation was reliably detectable and linearly increasing, to determine initial velocities. While there is a short delay before measurable TAL appears, this reflects the sensitivity threshold of LC-MS and transient intermediate kinetics, not a lack of catalysis. These clarifications have been incorporated into the figure legend of **Extended Data Fig. 4**.

The authors did not reanalyze the data of BktB acetyl-CoA. Normally, the highest resolution in crystallographic analysis should be set at $CC_{1/2}$ of 0.5 or better. The resolution for this structure should be around 2.5–2.3 Å?

Response: We thank the reviewer for this comment. The most commonly used $CC_{1/2}$ cutoff values (Karplus P.A. & Diederichs K. (2012). *Science*. **336**, 1030–1033.; Karplus P.A. & Diederichs K. (2015). *Curr. Opin. Struct. Biol.* **34**, 60–68.) for the highest resolution limit are 0.3 or 0.5, as described in the manual for the merging and scaling program Aimless (<https://cloud.ccp4.ac.uk/manuals/html-taskref/doc.task.Aimless.html#sd-correction>).

For the datasets in this study, we selected high resolution shell $CC_{1/2}$ cutoff values greater than 0.3, $\langle I/\sigma \rangle$ values greater than 1, and took into account other considerations such as the diffraction limit of the detector. For 3 of the datasets this led to $CC_{1/2}$ values greater than 0.5 (see table below). However, for the BktB acetyl-CoA dataset we used a $CC_{1/2}$ value of 0.3 as a cutoff with an $\langle I/\sigma \rangle$ less than 1, in order to provide as much map detail as possible given the relatively small size of the acetyl moiety.

	btkB – apo-enzyme	btkB – butyryl-CoA	btkB – acetyl-CoA	btkB – acetoacetyl-CoA
Highest Resolution (Å)	2.2	2.1	2.23	2.45
$CC_{1/2}$	0.673	0.558	0.309	0.773
I/σ	1.5	1.3	0.8	2.2

Extended Data Fig. 9d. Should "time (min⁻¹)" be Should "time (min)"?

Response: Thank you for pointing this out. We have corrected the labeling error in **Extended Data Fig. 9d**; the x-axis now correctly reads “time (min).”

REVIEWERS' COMMENTS

Reviewer #1 (Remarks to the Author):

Gibbs free energy change of Bktb step can be easily estimated from eQuilibrator 3.0

<https://equilibrator.weizmann.ac.il/>

Please check the attached file containing the Gibbs free energy change of BktB

Equivalently, search by the KEGG metabolite number, then calculate from eQuilibrator 3.0

Both reactions give the Gibbs free energy change of Bktb at the standard conditions ($\Delta_r G^\circ$) and physiologically relevant conditions ($\Delta_r G^m$) are + 25.0 KJ/mol, which all represents the non-decarboxylative Claisen reaction. Given the misleading information provided by the authors, I recommend a rejection of the current manuscript.

While the authors cited Tan's paper (Nature Catalysis 2020, A polyketoacyl-CoA thiolase-dependent pathway for the synthesis of polyketide backbones), Tan's paper also provided the misleading Gibbs free energy change of Bktb step.

Response: eQuilibrator uses KJ/mol unit, while Kcal/mol unit is in this work. 25.0 KJ/mol is equivalent with 5.97514 Kcal/mol, which is not very different from our estimated $\Delta_r G^m$ of the first non-decarboxylative condensation step between two acetyl-CoAs (7.06007 Kcal/mol). The difference of the estimates is probably due to the differences of estimation algorithm details used by eQuilibrator and Metacyc. These algorithms, though all based on the group contribution method, can vary in different tools, so it is understandable that estimation results could be different in Tan's paper, our work and eQuilibrator. Because eQuilibrator fails to estimate $\Delta_r G^m$ of TAL formation, we used $\Delta_r G^\circ$ of each reactant estimated by Metacyc to calculate the $\Delta_r G^m$, as described in the legend of Supplementary Figure 3. As long as estimation methods and results are accountable with reasonable references, it is probably not fair to rate them as 'misleading'. Moreover, all estimated standard Gibbs free energies are just estimates under standard conditions, so they are different from real Gibbs free energies under real experimental conditions. All the conclusions in this work were based on real experimental results, and estimated Gibbs free energy change was only used to hypothesize that TAL production from acetyl-CoA can occur due to estimated overall

thermodynamic favorability, which was then demonstrated by the *in vivo* and *in vitro* TAL production tests from acetyl-CoA.

Reviewer #2 (Remarks to the Author):

The authors revised the paper appropriately according to the comments made by the referee.

Response: Thanks for the appreciation of our work.

ROUND 3 REVIEWER 1 ATTACHMENT:

Gibbs free energy change of Bktb step can be easily estimated from eEquilibrator 3.0
<https://equilibrator.weizmann.ac.il/>

Reaction Gibbs Energy		
Estimated $\Delta_r G^m$	25.0 \pm 1.7 [kJ/mol]	
Estimated $\Delta_r G^\circ$	25.0 \pm 1.7 [kJ/mol] $K'_{eq} = 4.2 \times 10^{-5}$	
Catalyzed by	acetyl-CoA C-acetyltransferase [EC 2.3.1.9]	
pH	<input type="text" value="7.5"/>	<input type="range"/>
pMg	<input type="text" value="3.0"/>	<input type="range"/>
Ionic strength	<input type="text" value="0.25"/> M	<input type="range"/>

Equivalently, search by the KEGG metabolite number, then calculate from eEquilibrator 3.0

Reaction Gibbs Energy		
Estimated $\Delta_r G^m$	25.0 \pm 1.7 [kJ/mol]	
Estimated $\Delta_r G^\circ$	25.0 \pm 1.7 [kJ/mol] $K'_{eq} = 4.2 \times 10^{-5}$	
Catalyzed by	acetyl-CoA C-acetyltransferase [EC 2.3.1.9]	
pH	<input type="text" value="7.5"/>	<input type="range"/>
pMg	<input type="text" value="3.0"/>	<input type="range"/>
Ionic strength	<input type="text" value="0.25"/> M	<input type="range"/>

Both reactions give the Gibbs free energy change of Bktb at the standard conditions ($\Delta_r G^\circ$) and physiologically relevant conditions ($\Delta_r G^m$) are + 25.0 KJ/mol, which all represents the non-decarboxylative Claisen reaction. Given the misleading information provided by the authors, I recommend a rejection of the current manuscript.

While the authors cited Tan's paper (Nature Catalysis 2020, A polyketoacyl-CoA thiolase-dependent pathway for the synthesis of polyketide backbones), Tan's paper also provided the misleading Gibbs free energy change of Bktb step.